

# Tornado-Scale Vortices in the Tropical Cyclone Boundary Layer: Numerical Simulation with WRF-LES Framework

Liguang Wu[1,2], Qingyuan Liu[1] and Yubing Li[1]
[1]Pacific Typhoon Research Center and Key Laboratory of Meteorological Disaster of
Ministry of Education, Nanjing University of Information Science and Technology,
Nanjing, China
[2]Department of Atmospheric and Oceanic Sciences and Institute of Atmospheric
Sciences, Fudan University, Shanghai, China


Corresponding author address: Dr. Liguang Wu
Pacific Typhoon Research Center
Nanjing University of Information Science and Technology, Nanjing, Jiangsu 210044
E-mail: liguang@nuist.edu.cn





**Abstract**

The tornado-scale vortex in the tropical cyclone (TC) boundary layer (TCBL) has been
observed in intense hurricanes and the associated intense turbulence poses a severe threat
to the manned research aircraft when it penetrates hurricane eyewalls at a lower altitude.
In this study, a numerical experiment in which a TC evolves in a large-scale background
over the western North Pacific is conducted using the Advanced Weather Research and
Forecast (WRF) model by incorporating the large eddy simulation (LES) technique. The
simulated tornado-scale vortex shows the similar features as revealed with the limited
observational data, including the updraft/downdraft couplet, the sudden jump of wind
speeds, the favorable location, and the horizontal scale. It is suggested that the WRF-LES
framework can successfully simulate the tornado-scale vortex with the grids at the
resolution of 37 m that cover the TC eye and eyewall.
The simulated tornado-scale vortex is a cyclonic circulation with a small horizontal
scale of ~1 km in the TCBL. It is accompanied by strong updrafts (more than 15 m s$^{-1}$) and
large vertical components of relative vorticity (larger than 0.2 s$^{-1}$). The tornado-scale vortex
favorably occurs at the inner edge of the enhanced eyewall convection or rainband within
the saturated, high-$\theta_e$ layer, mostly below the altitude of 2 km. Nearly in all the simulated
tornado-scale vortices, the narrow intense updraft is coupled with the relatively broad
downdraft, constituting one or two updraft/downdraft couplets or horizontal rolling
vortices, as observed by the research aircraft. The presence of the tornado-scale vortex also
leads to significant gradients in the near surface wind speed and wind gusts.





## 1. Introduction

Tropical cyclones (TCs) pose a severe risk to life and property in TC-prone areas and the risk will increase due to the rapidly rising coastal population and buildings (Pielke et al. 2008; Zhang et al. 2009). One of the major TC threats is damaging winds. Uneven damage patterns often show horizontal scales ranging from a few hundred meters to several kilometers (Wakimoto and Black 1994; Wurman and Kosiba 2018), suggesting that TC threats are associated with both sustained winds and gusts. The latter are believed to result from small-scale coherent structures in the TC boundary layer (Wurman and Winslow 1998; Morrison et al. 2005; Lorsolo et al. 2008; Kosiba et al. 2013; Kosiba and Wurman 2014). The small-scale coherent structures may have significant implications for the vertical transport of energy in TCs and thus TC intensity and structure (Zhu 2008; Rotunno et al. 2009; Zhu et al. 2013; Green and Zhang 2014, 2015; Gao et al. 2017). While understanding of the coherent structure is very important for mitigating TC damage and understanding of TC intensity and structure changes, by now direct in situ observation and remote sensing measurements can only provide very limited information.

In the TC boundary layer (TCBL), observational analyses suggest that horizontal streamwise roll vortices prevail with sub-kilometer to multi-kilometer wavelengths (Wurman and Winslow 1998; Katsaros et al. 2002; Morrison et al. 2005; Lorsolo et al. 2008; Ellis and Businger 2010; Foster, 2013). Studies found that the rolls can result from the inflection point instability of the horizontal wind profiles in the TCBL (Foster 2005; Gao and Ginis 2014) and have significant influences on the vertical transport of energy in TCs (Zhu 2008; Rotunno et al. 2009; Zhu et al. 2013; Green and Zhang 2014, 2015; Gao et al. 2017). The TCBL is known to play a critical role in transporting energy and



controlling TC intensity (Braun and Tao 2000; Rotunno et al. 2009; Smith and
Montgomery 2010; Bryan 2012; Zhu et al. 2013; Green and Zhang 2015).

Another important small-scale feature is the so-called eyewall vorticity maximum

(EVM) (Marks et al. 2008) or tornado-scale vortices in the TCBL (Wurman and Kosiba
2018; Wu et al. 2018). So far, our understanding is mainly from a few observational
analyses based on limited data collected during the research aircraft penetration of
hurricane eyewalls. A WP-3D research aircraft from National Oceanic and Atmospheric
Administration (NOAA) encountered three strong updraft-downdraft couplets within one
minute while penetrating the eyewall of category 5 Hurricane Hugo (1989) at 450-m
altitude (Marks et al. 2008). The severe turbulence caused the failure of one of the four
engines and the people aboard were at a severe risk. The aircraft finally escaped with the
help of a U. S. Air Force reconnaissance WC-130 aircraft, which found a safe way out
through the eyewall on the northeast side of Hugo. Since then the aircraft mission has been
prohibited in the boundary layer of the TC eyewall. Later analysis indicated that the
dangerous turbulence was associated with a tornado-scale vortex, which is comparable to
a weak tornado in terms of its diameter of about 1 km and the estimated peak cyclonic
vorticity of 0.125 s$^{-1}$ (Marks et al. 2008). Such strong turbulence was also observed in
Hurricanes Isabel (2003) and Felix (2007) at different altitudes (Aberson et al. 2006;
Aberson et al. 2017). Understanding of the structure and evolution of the tornado-scale
vortex is hampered since it is difficult to directly observe due to its small horizontal scale
and the associated severe turbulence.

With advances in numerical models and computational capability, the large eddy

simulation (LES) technique has been incorporated into the Advanced Weather Research



and Forecast (WRF) model (Mirocha et al. 2010) and an increasing number of TC
simulations have been conducted with horizontal grid spacing less than 1 km (Zhu 2008;
Rotunno et al. 2009; Bryan et al. 2014; Stern and Bryan 2014; Rotunno and Bryan 2014;
Green and Zhang 2015). In LES the energy-producing scales of 3-dimensional (3D)
atmospheric turbulence in the planetary boundary layer (PBL) are explicitly resolved,
while the smaller-scale portion of the turbulence is parameterized (Mirocha et al. 2010).
Effort has been made to simulate the structure of the TC PBL eddies and the associated
influence on TC intensity. Zhu (2008) simulated the structure of the coherent large eddy
circulations and the induced vertical transport using the WRF-LES framework with
horizontal resolutions of 300 m and 100 m. When the horizontal resolution was decreased
from 185 to 62 m on the f-plane, Rotunno et al. (2009) found a sharp increase in randomly
distributed small-scale turbulent eddies, while 1-minute mean TC intensity began to
decrease. Green and Zhang (2015) performed several 6-hour one-way simulations of
Hurricane Katrina (2005) without a boundary layer parameterization (horizontal
resolutions of 333, 200, and 111 m). Rotunno et al. (2009) and Green and Zhang (2015)
suggest that the horizontal resolution should be below 100 m to simulate the development
of 3D turbulent eddies in TCBL.

It is clear that understanding of the tornado-scale vortex would enhance the safety of

flights into very intense TCs. In addition, the tornado-scale vortex may be responsible for
TC intensification by mixing the high-entropy air in the eye into the eyewall (Persing and
Montgomery 2003; Montegomery et al. 2006; Aberson et al. 2006) and track fluctuations
(Marks et al. 2008; Aberson et al. 2017). By simulating the tornado-scale vortex in the
TCBL, this study will particularly focus on the spatial distribution of the occurrence of the



tornado-scale vortex and the features of its 3D structures.
**2.    The numerical experiment**

In this study the numerical simulation is conducted using version 3.2.1 of the WRF

model. Following Wu and Chen (2016), two steps were taken to construct the initial
conditions for the numerical experiment. A symmetric vortex was first spun up without the
environmental flow on an f-plane for 18 hours and then the vortex was embedded in the
large-scale background of Typhoon Matsa (2005) from 0000 UTC 5 August to 1200 UTC
6 August. The large-scale environment was derived from the National Centers for
Environmental Prediction (NCEP) Final (FNL) Operational Global Analysis data with
resolution of $1.0° \times 1.0°$ using a 20-day low-pass Lanczos filter (Duchon 1979).

The spun-up vortex is initially located at the center of Typhoon Matsa (25.4°N,

123.0°E). The outermost domain centered at 30.0°N, 132.5°E covers an area of 6210×6210
$km^2$ with a horizontal spacing of 27 km. The numerical experiment is designed with six
two-way interactive domains embedded in the 27-km resolution domain to simulate
energetic 3-dimentional turbulent eddies in the TC eyewall and their influence on the TC
vortex, mesoscale rainbands and convective clouds. The horizontal spacing decreases by a
factor of 3 with the domain level. The corresponding horizontal resolutions are 9 km, 3 km,
1 km, 1/3 km (333 m), 1/9 km (~111 m) and 1/27 km (~37 m) and the numbers of their
grid meshes are 230×210, 432×399, 333×333, 501×501, 1351×1351, and 2431×2431,
respectively. The innermost domain covers the inner region of the simulated TC (90×90
$km^2$), including the eye and eyewall. Except the 27-km and 9-km resolution domains, the
other domains move with the TC. The model consists of 75 vertical levels (19 levels below



2 km) with a top of 50 hPa and is run over the open ocean with a constant sea surface
temperature 29ºC.

The physics options used in the simulation are as follows. The Kain-Fritsch cumulus

parameterization scheme and the WRF single-moment 3-class scheme are used in the
outermost domain (Kain and Fritsch 1993). The WRF 6-class scheme is selected in the
nested domains with no cumulus parameterization scheme (Hong and Lim 2006). The
Rapid Radiative Transfer Model (RRTM) and the Dudhia shortwave radiation scheme are
used for calculating long-wave radiation and shortwave radiation (Mlawer et al. 1997;
Dudhia 1989). The LES technique is used in the sub-kilometer domains (Mirocha et al.
2010) and the Yonsei University scheme is adopted for PBL parameterization in the other
domains (Noh et al. 2003).

The model is run for 36 hours and the 1/9-km-resolution and 1/27-km-resolution

domains are activated at 24 h. In the following analysis, we will focus on the hourly output
from 26 h to 36 h. The TC center is determined with a variational approach in which it is
located until the maximum azimuthal-mean tangential wind speed is obtained (Wu et al.
2006). A few variables are also stored at 3-second intervals during a 22-minute period from
the 30$^{th}$ hour.
**3. The simulated small-scale features**

The simulated TC takes a northern north west track (figure not shown). Figure 1 shows

its intensity in terms of the instantaneous and azimuthal maximum wind speeds at 10 m in
the 1/27 km-resolution domain. The instantaneous winds  are directly from the model
instantaneous output without any time averaging and the azimuthal wind speed is the wind
speed averaged azimuthally with respect to the TC center. The instantaneous maximum



wind speed fluctuates between 76.6 m s$^{-1}$ and 61.8 m s$^{-1}$ during the 12-hour period, while
the fluctuations in the azimuthal maximum wind speed is relatively small, ranging from
48.8 m s$^{-1}$ to 43.5 m s$^{-1}$. In particular, the TC maintains the azimuthal mean maximum wind
speed of ~45 m s$^{-1}$ after the innermost domain has been activated for two hours.
Figure 2a shows the simulated 500-m radar reflectivity at 27 h, indicating that the
eyewall is open to the south of the TC center. We examine the radar reflectivity field and
find that the opened eyewall persists during the 10-hour period. In addition, the location of
the enhanced convection relative to the TC center is generally steady. It is well known that
the eyewall asymmetry is associated with the vertical shear of the environmental flow
(Frank and Ritchie 2001, Braun and Wu 2007). In this study the vertical wind shear
calculated as the difference of wind vectors between 200 hPa and 850 hPa within a radius
of 300 km. As shown in the figure, the mean shear is 5.2 m s$^{-1}$ to the southeast over the 10-
hour period. In agreement with the previous studies, the enhanced eyewall reflectivity is
generally observed in the downshear left side. There are relatively small changes in the
RMW during the 11-hour period, ranging from 28.2 km to 30.7 km at 500 m.
Using the fine-scale dual Doppler data in the right front quadrant and eye of Hurricane
France (2004) as it made landfall on Florida, Kosiba and Wurman (2014) found linear
coherent structures with a wavelength of 400-500 m near the surface. Figure 2b shows the
simulated near-surface (10 m) wind speeds in the inner region at 27 h. The instantaneous
wind speed is dominated by quasi-linear coherent structures in the eyewall region. The
intense instantaneous wind speeds coincide with the enhanced eyewall convection shown
in Figure 2a. In order to show clearly the quasi-linear feature, we plot the instantaneous
wind speed in an area of 7×10 km$^2$ at this time (Fig. 3a). The small area is located in the





eyewall to the east of the TC center (Fig. 2b). The streaks of alternating high and low wind
speeds can be clearly seen, which are roughly aligned with the TC-scale flow with an
outward angle. We can see that the instantaneous wind speed exhibits large gradients across
the quasi-linear structures.

Figure 3b shows the perturbation wind filed at 500 m in the small area. The perturbation

winds are obtained by subtracting an 8-km moving mean. We compare the perturbation
winds with different sizes of the moving window. While the perturbation wind fields are
very similar, the wind speeds generally increase with the increasing window size. When
the wind size is larger than 8 km, there is little change in the perturbation wind speed. The
results are similar to those by subtracting the symmetric and wavenumber 1-3 components
with respect to the TC center. In the perturbation wind field, we can see two small-scale
cyclonic circulations. The most distinct one has a diameter of ~2 km. In the next section,
the two cyclones are identified as two tornado-scale vortices (M2701 and M2705).
Compared to Figure 3a, the two tornado-scale vortices also correspond to enhanced wind
speeds at 10 m.
**4.  Identification of EVMs**

As mentioned in Section 1, analyses of a few real cases in Atlantic intense hurricanes

indicate that the tornado-scale vortex is a small-scale feature that occurs in the turbulent
TC boundary, with vertical motion and relative vorticity extremes. Aberson et al. (2006)
and Aberson et al. (2016) analyzed the extreme updrafts in Hurricanes Isabel (2003) and
Felix (2007) and suggested that the strong updrafts were likely associated with the tornado-
scale vortex. The updraft of 25 m s$^{-1}$ in Isabel was detected by a GPS dropwindsonde just
below 800 hPa, while the updraft of 31 m s$^{-1}$ in Hurricane Felix (2007) was observed at the





flight altitude (~ 3 km). Marks et al. (2008) found that the tornado-scale vortex in Hurricane
Hugo (1989) was associated with a maximum vertical motion of 21 m s$^{-1}$ and a maximum
relative vorticity of 0.125 s$^{-1}$ at the altitude of 450 m. Based on these studies, the tornado-
scale vortex in the simulated TC is defined as a small-scale cyclonic circulation with the
diameter of 1-2 km below the altitude of 3 km, containing maximum upward motion larger
than 20 m s$^{-1}$ and maximum relative vorticity larger than 0.2 s$^{-1}$. The grid points that satisfy
the thresholds of vertical motion and relative vorticity belong to the same tornado-scale
vortex if they are within a distance of 1 km in the horizontal or vertical direction. We detect
the tornado-scale vortices using the output at one-hour intervals from 26 h to 36 h.

There are 24 tornado-scale vortices identified in the 10-hour output (Table 1). In the

table, the tornado-scale vortex is named with four digits. While the first two digits indicate
the hours of the simulation, the last two digits is the series number at the same hour. There
are four tornado-scale vortices with the maximum vertical motion more than 30 m s$^{-1}$ and
the maximum vertical component of relative vorticity larger than 0.4 s$^{-1}$. Except for the two
tornado-scale vortices at 36 h, the others occur during 26 h-31 h with 10 cases at 27 h. The
lull period is coincident with relatively weaker instantaneous maximum wind speed at 10
m although there is little difference in the azimuthal mean maximum wind speed (Fig. 1).
Examination indicates that the 10-m instantaneous wind speed maximum at 27 h is
associated with M2701. It is suggested that the tornado-scale vortex can lead to the
strongest wind gust in a TC.

Previous studies argued that the presence of the mesovortices intensifies the TC by

mixing the high-entropy air in the eye into the eyewall (Persing and Montgomery 2003;
Montegomery et al. 2006; Aberson et al. 2006). As shown in Figure 1, the azimuthal



maximum wind speed does not show any jump at 27 h, when there are 10 identified
tornado-scale vortices. In the following discussion, we will show that the mixing indeed
exits, but its effect on the azimuthal maximum wind speed cannot be detected. It is similar
with the conclusion from idealized numerical experiments conducted by Bryan and
Rotunno (2009). In fact, the azimuthal maximum wind speed (~45 m s$^{-1}$) is rather steady
during the 10-hour period after the innermost domain has been activated for two hours.

The number of the identified tornado-scale vortices is sensitive to the threshold of

vertical motion. If we relax the threshold of maximum vertical motion to 15 m s$^{-1}$, we can
identify 89 tornado-scale vortices during the 10-hour period (Fig. 4a). Nearly all the
tornado-scale vortices still occur in the same semicircle of the enhanced eyewall
reflectivity. The duration of the tornado-scale vortex is examined in the 3-second output.
The duration is counted as the consecutive period during which the maximum vertical
motion and relative vorticity are not less than the thresholds. For the thresholds of 20 m s$^{-1}$
in vertical motion and 0.2 s$^{-1}$ in relative vorticity, the mean duration is 40 seconds and the
longest is 138 seconds. We can conclude that the identified tornado-scale vortices are not
repeatedly counted in the 1-hour output.
**5. Spatial distribution of tornado-scale vortices**

Figure 4a shows the location of the maximum vertical motions of the detected tornado-

scale vortices. In this figure, we also plot the locations of the 89 tornado-scale vortices
identified with the threshold of maximum vertical motion of 15 m s$^{-1}$. The tornado-scale
vortices exclusively occur in the semicircle with intense convection from the east to the
northwest (Fig. 2a). Nearly all of the identified cases occur in the inward side of the radius
of maximum wind (RMW) or close to RMW, with two exceptions that are located outside



of the RMW (Fig. 4a). One is M2901, which is 11.8 km from the RMW, and the other is
M3601 being 7.3 km from the RMW (Table 1). Close examination indicates that the two
tornado-scale vortices occur between two high reflectivity bands.

Although the real tornado-scale vortices were observed by chance, they were also

associated with the intense radar reflectivity within the hurricane eyewall and sharp
horizontal reflectivity gradients (Aberson et al. 2006, Marks et al. 2008 and Aberson et al.
2016).  In agreement with these studies, all of the simulated tornado-scale vortices are
associated with sharp horizontal reflectivity gradients and most of them occur in the inner
edge of the intense eyewall convection within the RMW. As shown in Figure 2a, all of the
10 cases at 27 h are located in the inner edge of the intense reflectivity. It is suggested that
the tornado-scale vortex favorably occurs at the inner edge of the intense eyewall
convection.

Using the smoothed fields, we also calculate the Richardson number for each tornado-

scale (Table 1). It is calculated at each level and then averaged over a layer between 200
m and 800 m within a radius of 1.5 km from the location of the maximum vertical motion.
The Richardson number is small, and it is negative for seven cases. As suggested by Stern
et al. (2016), the strong updraft is mainly within a kilometer of the surface and it is
implausible for buoyancy to be the primary mechanism for vertical acceleration. In Figure
4a, the Richardson number is also plotted, which is averaged over the 10-hour period. We
can see that the tornado-scale vortices generally occur in the areas with the Richardson
number less than 0.25. The areas coincide with the semicircle of the enhanced eyewall
convection. Figure 4b further shows the field of the Richardson number at 27 h. The 10
tornado-scale vortices are all in an environment with the Richardson number less than 0.1.



Since the Richardson number is calculated as the ratio of the moist static stability to the
vertical wind shear in the TCBL, we speculate that the strong vertical wind shear in the
inward side of the intense eyewall convection is an important factor for the development
of tornado-scale vortices.

Figure 5 shows the vertical cross sections of tangential wind, radial wind, vertical

motion, reflectivity and relative vorticity below 2.5 km, which are averaged in the northeast
quadrant over the 10-hour period. Note that the radial locations of M2901 and M3601 are
not shown in Figure 5 due to the effect of the limited innermost domain on the calculation
of the azimuthal mean. Note that there are relatively small changes in the RMW during the
10-hour period. The maximum vertical motions associated with the tornado-scale vortices
are located inside the tilted RMW between the altitudes of 300 m and 1300 m.  Most of
them (71%) are found between 400 m and 600 m. The altitudes of the maximum vertical
motions generally increase when the inflow layer deepens outward. Figures 5b and 5c
further indicate that the tornado-scale vortices are generally found in the region of strong
vertical motion averaged over the northeastern quadrant, where the vortices are detected,
and large relative vorticity with sharp horizontal reflectivity gradient on the inward side of
the eyewall.
**6.    Tornado-scale vortex structure**

Using the high-resolution model output, we can explore the structural features of the

simulated tornado-scale vortex. After examination of all of the identified 24 tornado-scale
vortices, we find that they can be classified into three categories in structure.





The first category includes 17 cases, accounting for 71% of the total. Their structural
features can be represented by M2701, one of the four strongest tornado-scale vortices,
located 4.3 kilometers from the 500-m RMW inward (Table 1). In fact, the four strongest
belong to the same category. In this category, nearly all of the maximum vertical motions
occur around the altitude of 500 m, except M3001. The maximum vertical motion of
M2701 is 31.98 m s$^{-1}$ at the altitude of 400 m, while the maximum relative vorticity of 0.55
s$^{-1}$ occurs at 200 m (Table 1). The 3D structure of the tornado-scale vortex can be clearly
demonstrated by the streamlines of perturbation winds near the strong updraft (Fig. 6). The
flows curl cyclonically upward from the surface (Fig. 6a). The tornado-scale vortex is
manifested by a small-scale circulation extending upward to ~1.5 km. Besides, the tornado-
scale vortex is closely associated with horizontal rolls (Fig. 6b).
Figure 7 shows the vertical cross section of vertical motion, equivalent potential
temperature, and simulated radar reflectivity along the line in Figure 3b for M2701. The
inflow from the outward side and the outflow from the eye side converge near the surface
to the strong updraft that is below ~1.5 km. The updraft and the downward motion to its
radially outward frank constitute a horizontal rolling vortex. On the top of the updraft, there
is a layer of the high equivalent potential temperature ($\theta_e$) layer (Fig. 7b). To the eye side
of the updraft, there is a high $\theta_e$ layer below ~1.5 km. The high $\theta_e$ layer tilts upward and
extends outward. The large radar reflectivity can be found below the high $\theta_e$ layer (Fig. 7c).
The intense updraft is located in the inner edge of the large radar reflectivity region. While
the large radar reflectivity is part of the eyewall, the high $\theta_e$ layer should be the indication
of the air in the eye. In addition, as suggested by Aberson et al. (2006) and Marks et al.





(2008), the strong updraft is within a saturated layer (Fig. 8a), coinciding with high relative
vorticity (Fig. 8c).
To the right of the updraft (Fig. 7b), another high $\theta_e$ layer can be seen at the altitude of
~500 m.  We check other cases in the category and find that the lower-altitude high $\theta_e$ layer
does not always present. The downward motion at ~500 m may be responsible for the
lower-altitude high $\theta_e$ layer. The relatively low $\theta_e$ near the surface corresponds to the inflow
layer, which brings lower $\theta_e$ into the updraft. The high $\theta_e$ air meets with the cold inflow air,
resulting in relatively lower $\theta_e$ in the strong updraft. It is indicated that the high $\theta_e$ air in
the eye is entrained into the TC eyewall.
Previous studies have shown that the quasi-linear bands are closely associated with the
horizontal rolls in the TC boundary due to the upward and downward momentum transports
(Wurman and Winslow 1998; Katsaros et al. 2002; Morrison et al. 2005; Lorsolo et al.
2008; Ellis and Businger 2010; Foster 2013). To demonstrate the relationship, Figure 8b
shows the cross section of winds along the line shown in Figure 2b and the corresponding
wind speeds at 10 m and 400 m. The figure clearly shows that the wind speed fluctuations
at 10 m are associated with the changes of the vertical motions. The wind speed jump is
significant across the intense updraft (Fig. 8b). At 10 m, the wind speed suddenly increases
from ~30 m s$^{-1}$ to ~65 m s$^{-1}$. Note that the wind speed jump is larger at 400 m, ranging from
~35 m s$^{-1}$ to ~90 m s$^{-1}$. Marks et al. (2008) reported that the wind speed at 450-m altitude
increased rapidly from <40 ms$^{-1}$ to 89 ms$^{-1}$ in the Hurricane Hugo (1989) when the NOAA
research aircraft encountered an EVM. While the downward motion is consistent with the
strong wind speed jumps, we argue that the superposition of the cyclonic circulation of the



tornado-scale vortex also play an important role in enhancing wind gusts on its radially
outward side.

There are three tornado-scale vortices in the second category, including M2706, M2707

and M2708. The structural features can be represented by M2708. In M2708, the maximum
vertical motion and relative vorticity occur at 900 m and 800 m, respectively (Table 1).
The vertical motion of more than 12 m s$^{-1}$ extends vertically from the near surface to ~2
km (Fig. 9a). In this category, we cannot see the warm air with high $\theta_e$ (Fig. 9b) and the
strong updraft is located in a statically unstable stratification (Table 1). The wind speed at
the altitude of 900 m varies by ~20 m s$^{-1}$ across the updraft, while the wind speed gradient
is relatively weak at 10 m (Fig. 9c).

The third category includes four cases: M2600, M2703, M2705 and M3002, in which

the updraft occurs in a statically stable stratification (Table 1). Here we use M3002 as an
example to show its vertical structure. As shown in Figures 10a, the updraft is elevated
between 0.5 km and 2 km. The maximum vertical motion and relative vorticity are found
at the altitude of 1300 m. In this category, a pronounced feature is the deep low $\theta_e$ (less
than 364 K) layer in the inflow layer (Fig. 10b). As shown in Figure 10c, the gradient of
the wind speed at 10 m is not clear while there is a speed jump of ~30 m s$^{-1}$ in the vicinity
of the updraft at 1300 m.
**7. Summary**

The tornado-scale vortex or EVM in the TCBL has been observed in intense hurricanes

and is always associated strong turbulence. To understand complicated interactions of the
large-scale background flow, TC vortex, mesoscale organization, down to fine-scale


turbulent eddies, a numerical experiment in which a TC evolves in a typical large-scale
background over the western North Pacific is conducted using the WRF-LES framework
with six nesting grids. The simulated tornado-scale vortex shows the similar features as
revealed with the limited observational data, including the updraft/downdraft couplet, the
sudden jump of the wind speed, the favorable location, and the horizontal scale. It is
suggested that the WRF-LES framework can successfully simulate the tornado-scale
vortex with the grids at the resolution of 37 m that cover the TC eye and eyewall.
Following Wu et al. (2018), the tornado-scale vortex can be defined as a small-scale
cyclonic circulation with the maximum vertical motion not less than 20 m s$^{-1}$ and maximum
relative vorticity not less than 0.2 s$^{-1}$. A total of 24 tornado-scale vortices can be identified
in the 10-hour output.  Nearly all of them are within or close to the RMW. Most of them
occur in the inward side of the intense eyewall convection, mostly below the altitude of 2
km. Tornado-scale vortices are mostly in neutral or stable stratification within the saturated,
high-$\theta_e$ layer.. The tornado-scale vortex generally occurs in the areas with the Richardson
number less than 0.25. We speculate that the strong vertical wind shear in the inward side
of the intense eyewall convection is an important factor for the development of tornado-
scale vortices.
The simulated tornado-scale vortex has a small horizontal scale of 1-2 km in the TCBL.
It is accompanied by strong updrafts (more than 15 ms$^{-1}$) and a cyclonic circulation with
large vertical components of relative vorticity (larger than 0.2 s$^{-1}$). The tornado-scale vortex
is closely associated with horizontal rolls. Nearly in all of the simulated tornado-scale
vortex cases, the narrow intense updraft is coupled with the relatively broad downdraft,
constituting an updraft/downdraft couplet or horizontal rolling vortex, as observed by the



research aircraft. Since the tornado-scale vortex is associated with intense updrafts and
strong wind gusts, its presence can pose a severe threat to the eyewall penetration of
manned research aircraft and the strong wind gusts associated with tornado-scale vortices
can pose a severe risk to coastal life and property.
**Acknowledgments.** We thank Prof. Ping Zhu of Florida International University for aiding
with the WRF-LES framework. This research was jointly supported by the National Basic
Research Program of China (2015CB452803), the National Natural Science Foundation of
China (41730961, 41675051, 41675009), and Jiangsu Provincial Natural Science Fund
Project (BK20150910). The numerical simulation was carried out on the Tianhe
Supercomputer, China.

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






**Table caption**
Table 1 List of the identified tornado-scale vortices in the TCBL with the maximum

updraft (m s$^{-1}$) and relative vorticity (s$^{-1}$) and the corresponding altitudes (m) in

the parentheses. The location column lists the radial distance from the TC center

and the relative distance to the 500-m radius of maximum wind in the parentheses.

The Richardson number (Ri) is averaged over the layer between 200-800 m

within a radius of 1.5 km. The four strongest EVMs are indicated in bold.







**Figure caption**
Figure 1 Intensity of the simulated tropical cyclone during 24-36 h in terms of
instantaneous (red) and azimuthal (blue) maximum wind speeds at 10 m.

Figure 2 Simulated radar reflectivity (dBZ) at 500 m (a) and wind speed (m s$^{-1}$) at 10
542         m (b)  within an area of 40×40 km$^2$ at 27 h. The plus signs and solid circles
indicate the TC center and the radius of maximum wind. The rectangle shows
the area used in Fig. 3a. The arrow shows the vertical wind shear of 7.0(27h)
545         m s$^{-1}$ between 200 hPa and 850 hPa.

Figure 3 (a) 10-m wind speed (m s$^{-1}$) and wind vectors and (b) the perturbation wind
vectors and vertical component of relative vorticity (shading) at 500 m in the
area shown in Fig. 2b. The straight line is the location of the vertical cross
section in Figure 7 and M2701 and M2705 are the two tornado-scale vortices in
the small area.

Figure 4 (a) Horizontal distribution of the tornado-scale vortices identified with the
thresholds of 15 m s$^{-1}$ (yellow dots) and 20 m s$^{-1}$ (red dots) in vertical motion
and the Richardson number (shading) averaged over 26-36 h; (b) the same as
(a), but for 27 h. The solid circle is the 500-km radius of maximum wind and
dashed circles indicate the distances from the TC center at 10-km intervals.

Figure 5 Vertical cross sections of (a) tangential (shading) and radial (contour,
interval: 2 m s$^{-1}$) wind speeds, (b) upward motion (contour, interval: 0.5 m s$^{-1}$)
and radar reflectivity (shading), and (c) tangential wind (contour, interval: 4 ms$^{-1}$)
$^{1}$) and the vertical component of relative vorticity (shading, unit: s$^{-1}$), which are
averaged over the northeastern quadrant during 26 h-36 h. The dots are the
locations of identified tornado-scale vortices. The dashed white lines indicate
the radius of maximum wind. The vertical and horizontal axes indicate the
altitude (km) from the surface and the relative distances (km) from the TC
center.





Figure 6 (a) The streamlines of the horizontal perturbation winds for M2701 and the

wind speed (shading) at the altitude of 10 m. (b) The vertical slice of the

perturbation winds for M2701. The warm (cold) color of the streamline

indicates the upward (downward) vertical velocity perturbation and the vectors

show the near-surface wind fields. The vertical and horizontal axes indicate the

altitude (km) from the surface and the relative distances (km) from the nearest

corner, respectively.

Figure 7 The vertical cross sections of the perturbation winds (vector) and (a)

vertical motion, (b) equivalent potential temperature, and (c) radar reflectivity

(shading) for M2701 along the line in Figure 3b. The abscissa indicates the

relative outward distance.

Figure 8 (a) the vertical cross section of perturbation winds (vector) and relative

humidity (shading) for M2701, (b) the 500-m (blue) and 10-m (black) wind

speeds and the 400-m relative vorticity for M2701 along the line in Figure 3b.

The abscissa indicates the relative outward distance.

Figure 9 The vertical cross sections of the perturbation winds (vector) and (a)

vertical motion, (b) equivalent potential temperature for M2708, and (c) the

corresponding 900-m (blue) and 10-m (black) wind speeds. The abscissa

indicates the relative outward distance.

Figure 10 The vertical cross sections of the perturbation winds (vector) and (a)

vertical motion, (b) equivalent potential temperature for M3002, and (c) the

corresponding 1300-m (blue) and 10-m (black) wind speeds. The abscissa

indicates the relative outward distance. The abscissa indicates the relative

outward distance.



Table 1 List of the identified tornado-scale vortices in the TCBL with the maximum updraft (m s$^{-1}$) and relative vorticity (s$^{-1}$) and the corresponding altitudes (m) in the parentheses. The location column lists the radial distance from the TC center and the relative distance to the 500-m radius of maximum wind in the parentheses. The Richardson number (Ri) is averaged over the layer between 200-800 m within a radius of 1.5 km. The four strongest EVMs are indicated in bold.

| No. | Updraft | Vorticity | Location | Ri |
|---|---|---|---|---|
| M2600 | 22.75(800) | 0.36(400) | 23.6 (-5.5) | 0.095 |
| M2601 | 22.39(600) | 0.23(500) | 25.3 (-3.8) | 0.111 |
| M2700 | 27.37(500) | 0.45(200) | 25.6 (-3.0) | 0.017 |
| **M2701** | **31.98(400)** | **0.55(200)** | **24.3 (-4.3)** | **-0.008** |
| M2702 | 21.40(300) | 0.30(300) | 21.1 (-7.5) | 0.029 |
| M2703 | 20.46(400) | 0.23(400) | 27.9 (-0.7) | 0.013 |
| M2704 | 27.76(500) | 0.34(400) | 22.8 (-5.8) | 0.032 |
| M2705 | 22.26(600) | 0.24(600) | 27.9 (-0.7) | 0.038 |
| M2706 | 20.93(600) | 0.23(500) | 20.7 (-7.9) | -0.031 |
| M2707 | 20.30(700) | 0.21(700) | 29.6 (1.0) | -0.011 |
| M2708 | 22.20(900) | 0.29(800) | 31.2 (2.6) | -0.037 |
| M2709 | 21.49(800) | 0.22(800) | 22.8 (-5.8) | 0.052 |
| M2800 | 20.12(400) | 0.23(400) | 27.0 (-1.7) | 0.030 |
| M2801 | 24.36(600) | 0.39(400) | 24.2 (-4.5) | -0.037 |
| M2802 | 22.14(600) | 0.30(500) | 29.0 (0.3) | 0.029 |
| M2803 | 20.14(500) | 0.23(500) | 26.6 (-2.1) | 0.025 |
| **M2900** | **34.98(400)** | **0.48(200)** | **27.5 (-1.7)** | **0.042** |
| M2901 | 20.95(400) | 0.21(400) | 41.0 (11.8) | 0.017 |
| **M3000** | **35.77(400)** | **0.48(300)** | **28.1 (-0.1)** | **0.044** |
| **M3001** | **38.33(900)** | **0.49(400)** | **27.7 (-0.5)** | **0.067** |
| M3002 | 21.43(1300) | 0.29(1300) | 29.8 (1.6) | 0.083 |
| M3100 | 20.87(600) | 0.24(700) | 25.1 (-3.3) | -0.106 |
| M3600 | 22.00(400) | 0.35(400) | 24.1 (-6.6) | 0.146 |
| M3601 | 22.68(600) | 0.23(500) | 38.0 (7.3) | -0.073 |






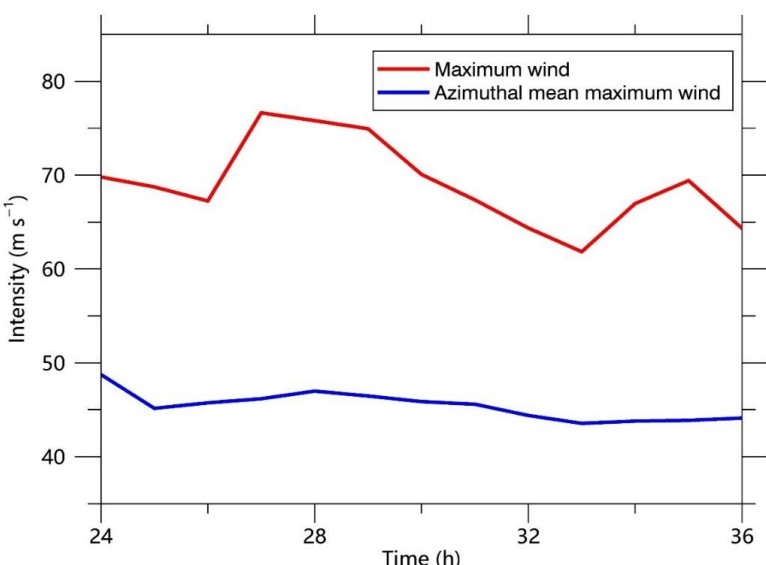


Figure 1 Intensity of the simulated tropical cyclone during 24-36 h in terms of
instantaneous (red) and azimuthal (blue) maximum wind speeds at 10 m.



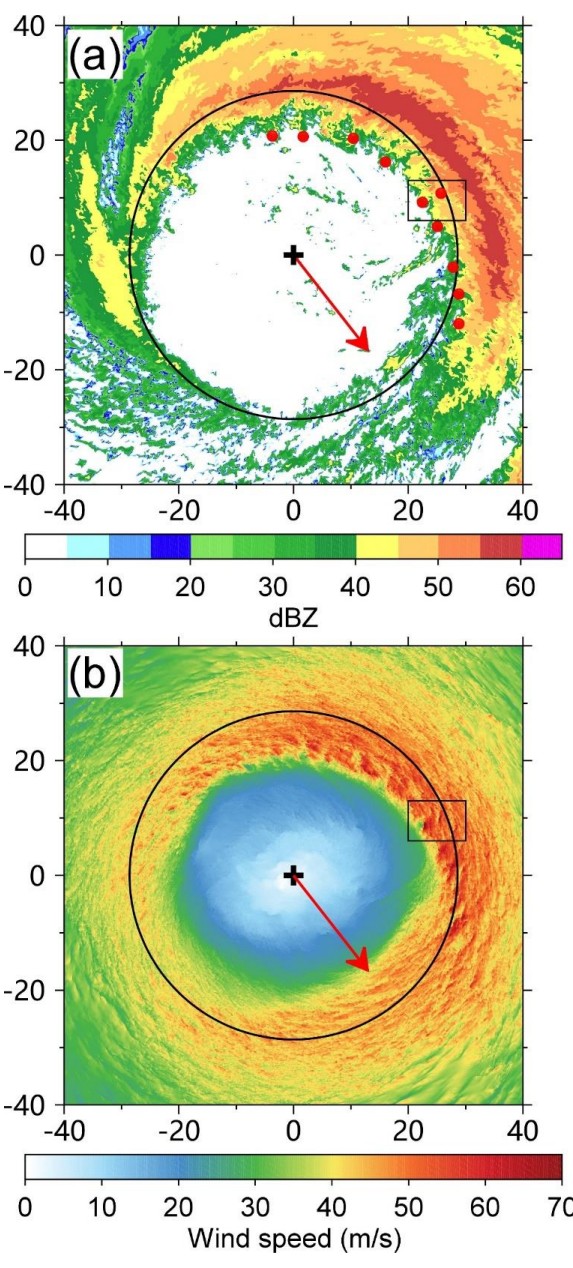


Figure 2 Simulated radar reflectivity (dBZ) at 500 m (a) and wind speed (m s$^{-1}$) at 10 m (b)
within an area of 40×40 km$^2$ at 27 h. The plus signs and solid circles indicate the TC center
and the radius of maximum wind. The rectangle shows the area used in Fig. 3a. The arrow
shows the vertical wind shear of 7.0(27h) m s$^{-1}$ between 200 hPa and 850 hPa.

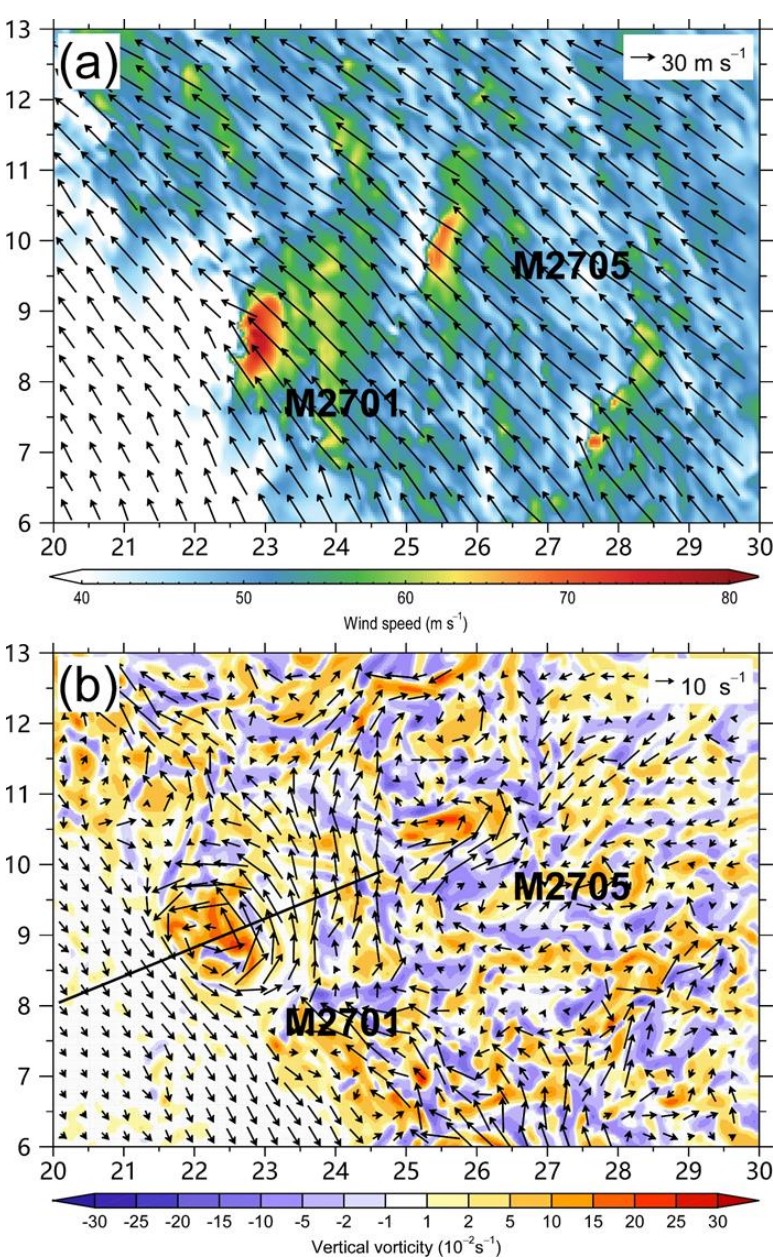

608

Figure 3 (a) 10-m wind speed (m s$^{-1}$) and wind vectors and (b) the perturbation wind vectors
and vertical component of relative vorticity (shading) at 500 m in the area shown in Fig.
2b. The straight line is the location of the vertical cross section in Figure 7 and M2701 and
M2705 are the two tornado-scale vortices in the small area.



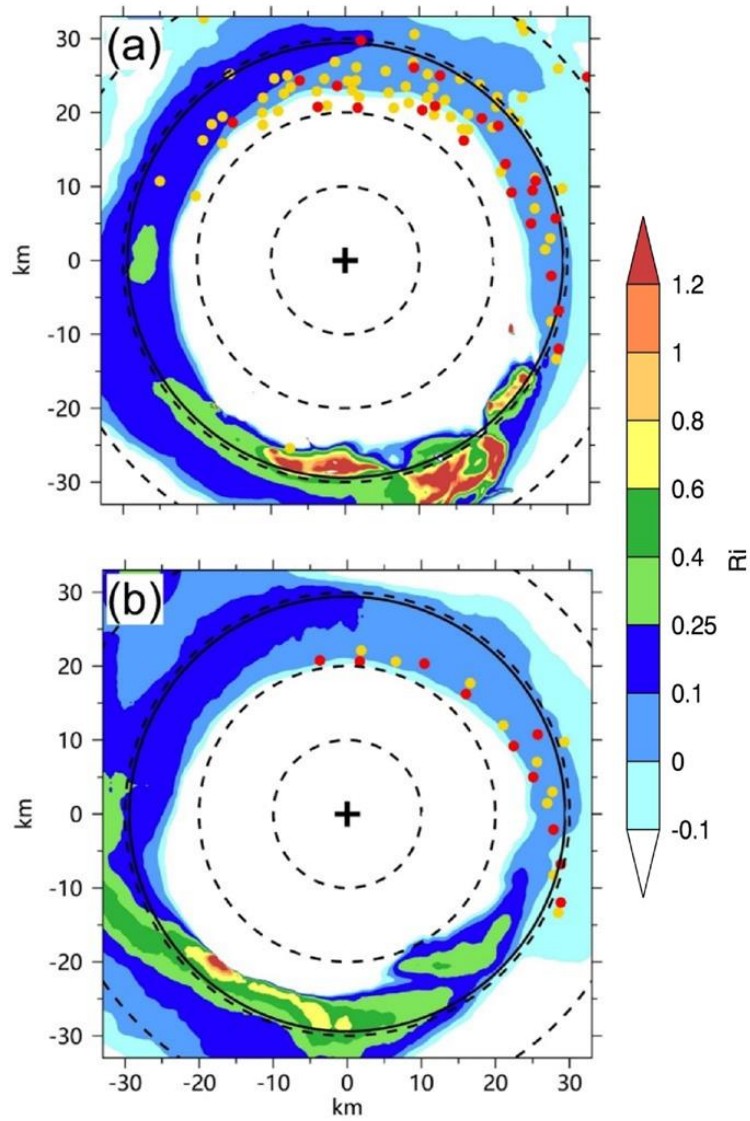

613

Figure 4 (a) Horizontal distribution of the tornado-scale vortices identified with the thresholds of 15 m s$^{-1}$ (yellow dots) and 20 m s$^{-1}$ (red dots) in vertical motion and the Richardson number (shading) averaged over 26-36 h; (b) the same as (a), but for 27 h. The solid circle is the 500-km radius of maximum wind and dashed circles indicate the distances from the TC center at 10-km intervals.





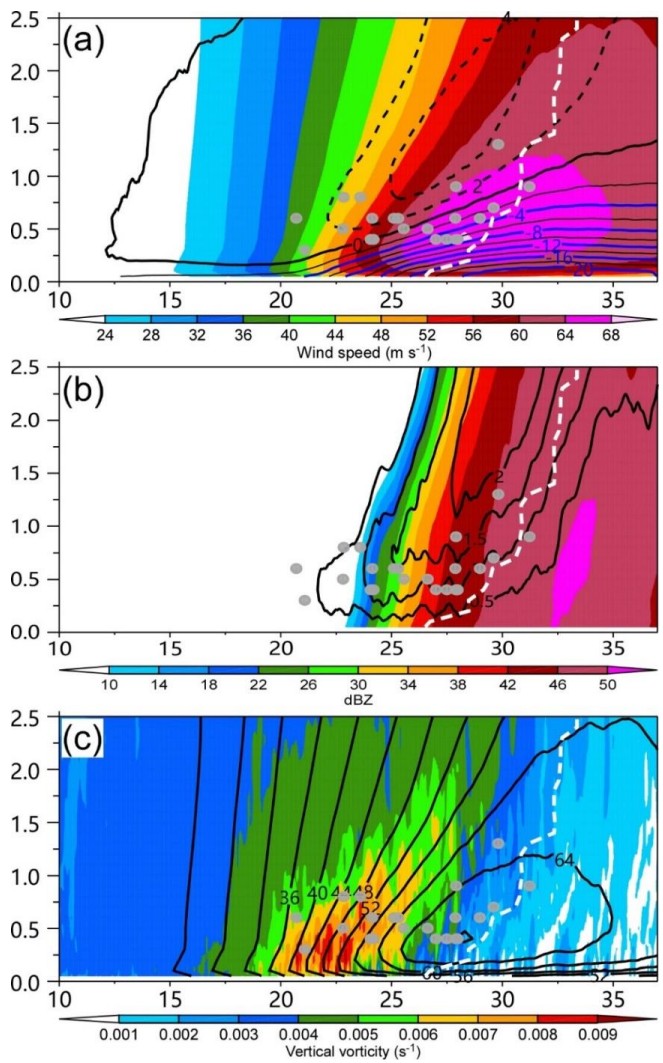

619

Figure 5 Vertical cross sections of (a) tangential (shading) and radial (contour, interval: 2 m s$^{-1}$) wind speeds, (b) upward motion (contour, interval: 0.5 m s$^{-1}$) and radar reflectivity (shading), and (c) tangential wind (contour, interval: 4 ms$^{-1}$) and the vertical component of relative vorticity (shading, unit: s$^{-1}$), which are averaged over the northeastern quadrant during 26 h-36 h. The dots are the locations of identified tornado-scale vortices. The dashed white lines indicate the radius of maximum wind. The vertical and horizontal axes indicate the altitude (km) from the surface and the relative distances (km) from the TC center.





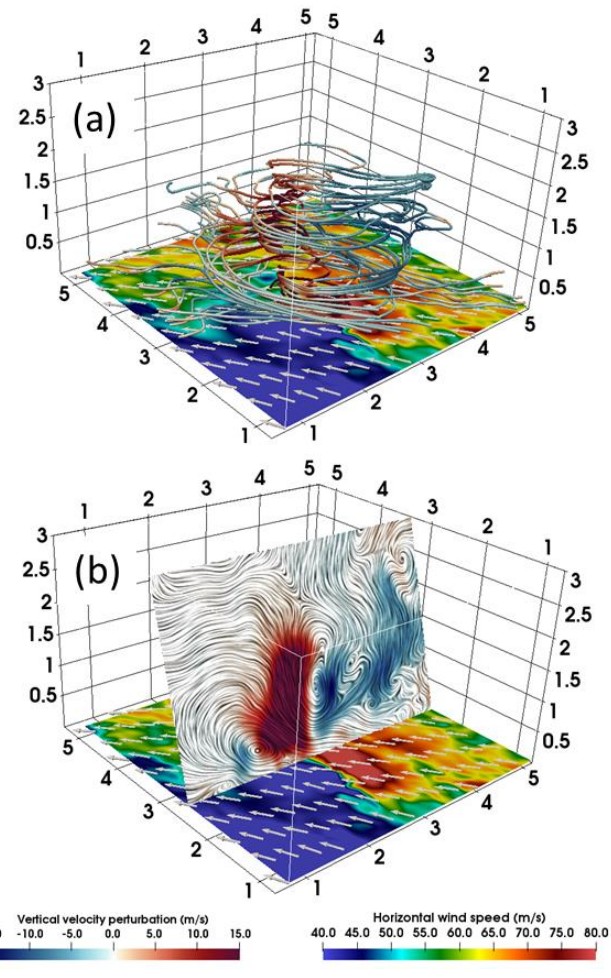

627

Figure 6 (a) The streamlines of the horizontal perturbation winds for M2701 and the wind
speed (shading) at the altitude of 10 m. (b) The vertical slice of the perturbation winds for
M2701. The warm (cold) color of the streamline indicates the upward (downward) vertical
velocity perturbation and the vectors show the near-surface wind fields. The vertical and
horizontal axes indicate the altitude (km) from the surface and the relative distances (km)
from the nearest corner, respectively.

634

635

636





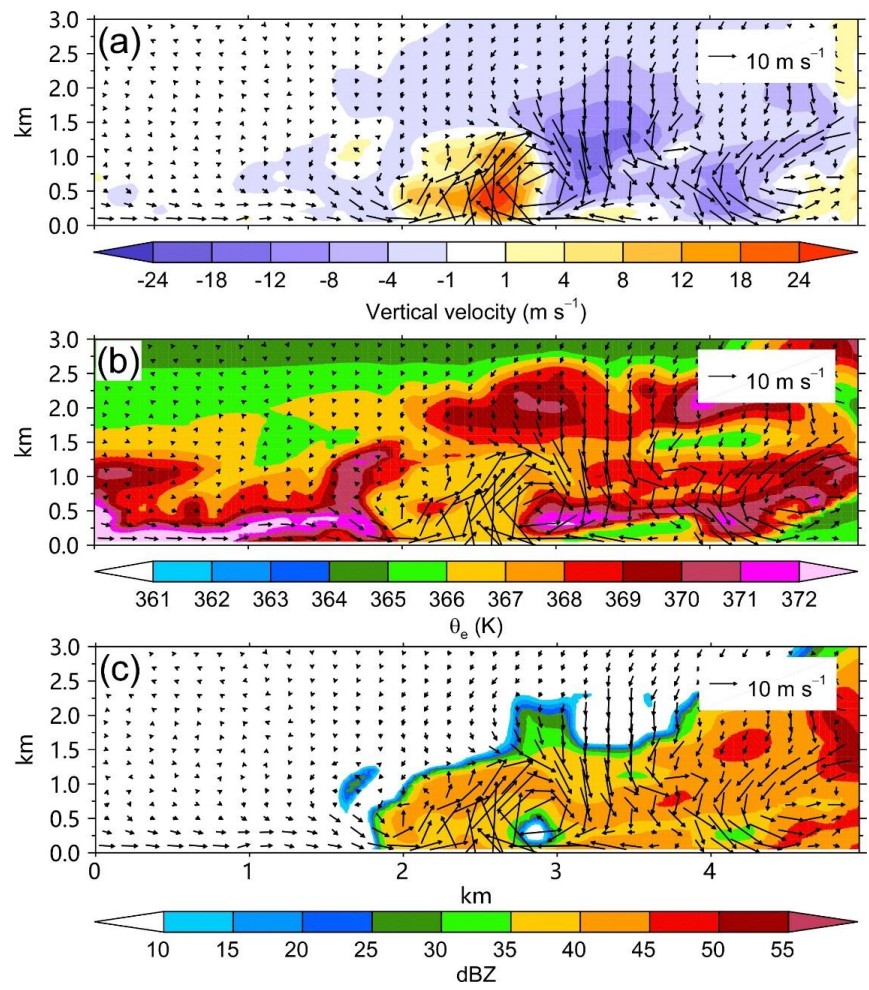

Figure 7 The vertical cross sections of the perturbation winds (vector) and (a) vertical motion, (b) equivalent potential temperature, and (c) radar reflectivity (shading) for M2701 along the line in Figure 3b. The abscissa indicates the relative outward distance.



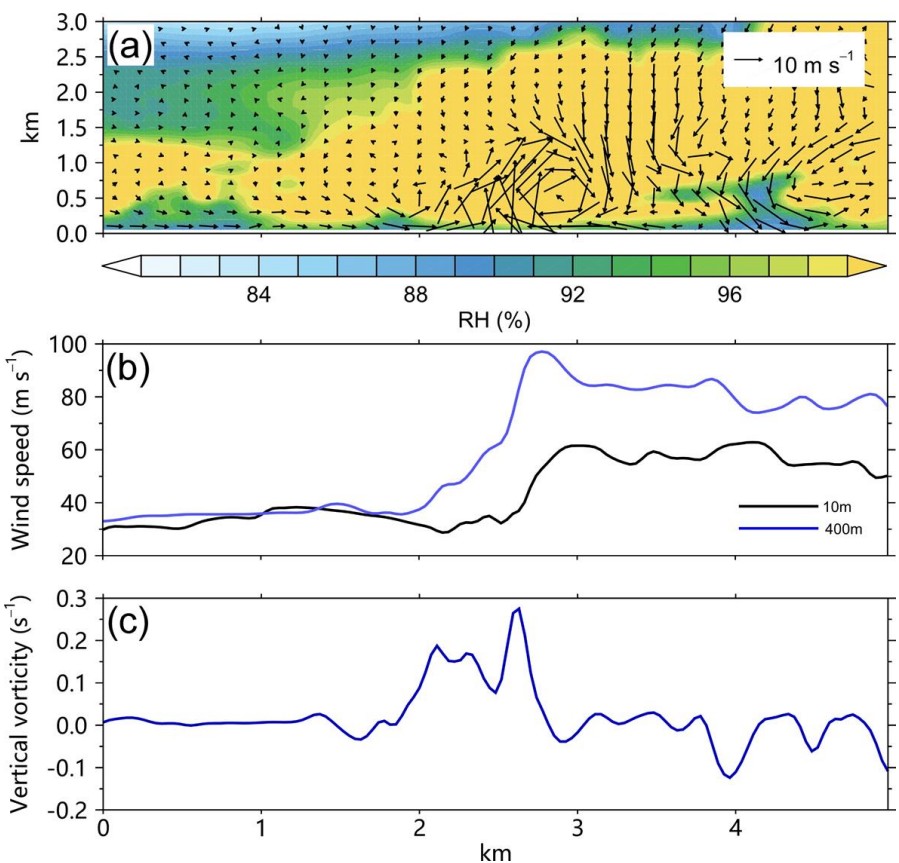

643

Figure 8 (a) the vertical cross section of perturbation winds (vector) and relative humidity
(shading) for M2701, (b) the 500-m (blue) and 10-m (black) wind speeds and the 400-m
relative vorticity for M2701 along the line in Figure 3b. The abscissa indicates the relative
outward distance.




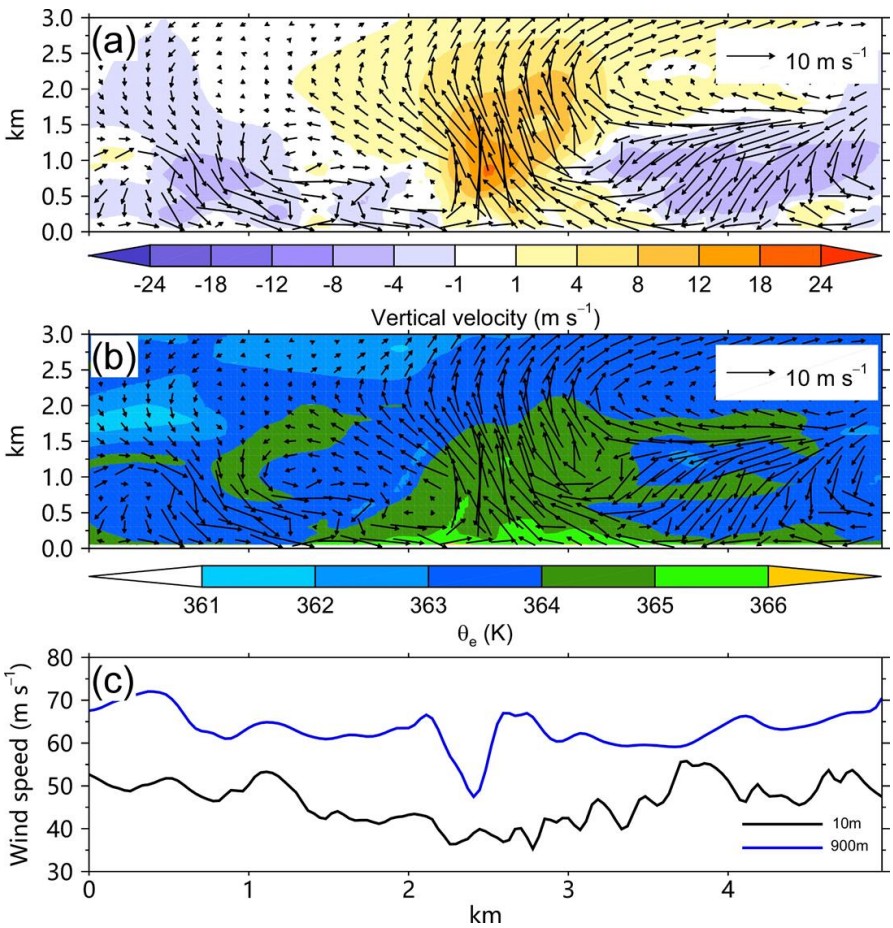


Figure 9 The vertical cross sections of the perturbation winds (vector) and (a) vertical
motion, (b) equivalent potential temperature for M2708, and (c) the corresponding 900-m
(blue) and 10-m (black) wind speeds. The abscissa indicates the relative outward distance.





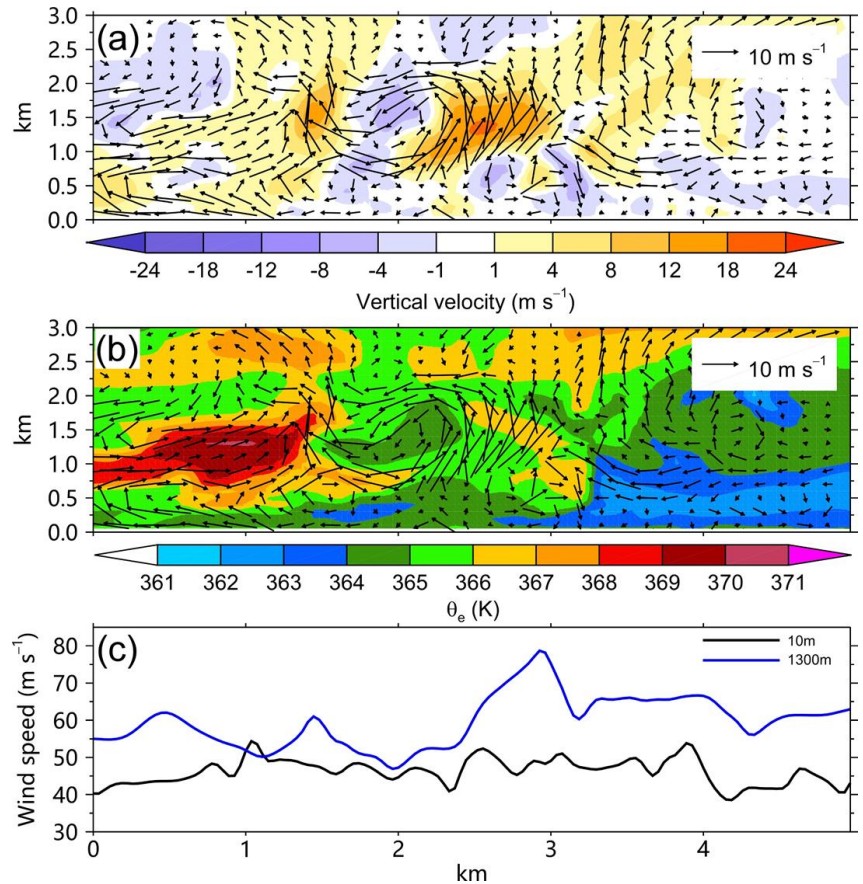


Figure 10 The vertical cross sections of the perturbation winds (vector) and (a) vertical
motion, (b) equivalent potential temperature for M3002, and (c) the corresponding 1300-
m (blue) and 10-m (black) wind speeds. The abscissa indicates the relative outward
distance. The abscissa indicates the relative outward distance.
