# Peer review of "Tornado-Scale Vortices in the Tropical Cyclone Boundary Layer: Numerical Simulation with WRF-LES Framework"

_Atmospheric Chemistry and Physics, 2018_

## Referee Comment (RC1) · Anonymous Referee #3 · 29 Oct 2018

This paper studies the tornado-scale vortex embedded in the tropical cyclone (TC) boundary layer (TCBL) by using very high horizontal resolution ($\sim$37m at inner most domain) of the Advanced Weather Research and Forecast (WRF) model with the large eddy simulation (LES) technique. The WRF-LES framework used in this study successfully simulated the tornado-scale vortex and showed the features similar to the observations, such as the updraft/downdraft couplet, the sudden jumps of wind speed at the inner side of the RMW below 2km altitude with scales of 1-2km. The study provides detailed insightful analysis on the small scale features, scale integrations in TC as many of the operational forecast centers moves their hurricane model system towards higher and higher resolution systems. It is amazing that TC dynamic model

is able to simulate such detailed small-scale features. This study provides insightful information for hurricane modelers in both research and operational communities to develop very high resolution TC forecast systems. 1. The paper uses the explicit convection for all nested domains (P7, line 149). However, the 9km resolution is still in the upper end of scale gray-zone between parameterized and explicit convection. 2. The time step used in this simulation is not mentioned in the paper? 3. The study raised an interesting question on the representation of TC intensity in terms of wind speed in the best track. Currently, for West Pacific TC, the JMA, along with most meteorological agencies around the world, uses a 10-minute averaging time for TC's maximum sustained wind, CAM uses 2-minutes while the JTWC uses a 1-minute averaging time. Ten-minute wind speeds are slower than 1-minute wind speeds, usually by a factor of 1.10–1.20 for tropical cyclones. In the paper, the authors showed that the life time of the tornado-scale vortex is around 40sec-138sec (PP11, line249-250), which seems indicating that 1-2 minutes averaging time is more adequate to represent max sustained wind in TC, especially for high resolution hurricane model verification. On the other hand, the "Wind-Pressure relationship" is commonly used in TC verification by comparing model produced "W-P relationship" with the one in best track. While the observed TC's minimum sea level pressure is not affected by tornado-scale vortex, the maximum sustained wind speed in best track needs to be more accurate when high resolution TC model is verified.

Minor corrections: 1. P5, line 108, "When the horizontal resolution was decreased . . ." should read "When the horizontal resolution was increased . . .: 2. P9, line 198, ". . .wind size. . ." should be ". . .window size . . .."
* * *

---

## Referee Comment (RC2) · Anonymous Referee #4 · 30 Oct 2018

The manuscript demonstrate that it is possible to successfully simulate tornado-scale vortices in WRF integrations set up as a Large Eddy Simulations, with 6 telescopic grids and the smallest grid point distance being 37m. Because the findings are roughly consistent with the limited observations, successful numerical simulations have the potential to be the most reliable source of turbulent scale statistics for coarser simulations. The study defines a criteria for finding tornado-scale structures and identifies several of them clarifying them into three categories based on structure. It is shown that the tornado-scale vortices are associated with horizontal rolls and it is speculated that strong vertical shear in the inward side of the eyewall convection is relevant in the development of tornado-scale vortices. Dipoles of updrafts and downdrafts and drastic

changes on wind speed are documented. Because the findings are roughly consistent with the limited observations, successful numerical simulations have the potential to be the most reliable source of turbulent scale statistics for coarser simulations.

The manuscript shows that model could simulate the tornado-scale vortices in TC boundary layer at inner eyewall region, also groups these tornado-scale vortices into three categories. It is important to study the small features of TC and impact of these features to TC intensity and structure. This manuscripts is well written. So, recommend minor revision.

Some comments and suggestions are provided below:

Line 92-94: "Such strong turbulence was also observed in Hurricane Isabel (2003) and Felix (2007) at different altitudes (Aberson et al. 2006; Aberson et al .2007)". It is better to list the exact altitudes of this "different altitudes" to make sure these are related to TC BL turbulence.

Line 94-96: "Understanding of the structure and evolution of the . . . . . . severe turbulence." This sentence doesn't match the logic. The reason to understanding of this small structure turbulence should be it is important for determining storm intensity, it should not be hard to observe. Using numerical simulation is because it is hard to observe.

Line 132-145: The finest resolution of horizontal resolution of this simulation is 37 meters, while the vertical resolution is only 75 levels. This concerns as the ratio of horizontal resolution and the vertical resolution could play a big role in the 3D simulations.

Line 156-157: "we will focus on the hourly output from 26h to 36h." Since this is tornado scale feature and the horizontal resolution reaches 37m, hourly output is too coarse and would miss some features. Suggest taking a more aggressive evaluation of output of the order of minutes (at least 15 minutes).

It is better to indicate the red dots as tornado-scale vortices in Fig.2a in figure caption.

[Figure]

Line 228: change "1ull" to "full"

---

## Referee Comment (RC3) · Anonymous Referee #5 · 1 Nov 2018

**Review** of "Tornado-Scale Vortices in the Tropical Cyclone Boundary Layer: Numerical Simulation with the WRF-LES Framework", by Liguang Wu, Qingyuan Liu, and Yubing Li.

**General Comments**:

In this study, the authors use WRF-LES to simulate a quasi-idealized tropical cyclone (in the environment of a real TC), for the purpose of investigating tornado-scale vortices. They find such vortices along the inner eyewall, concentrated in the left-of-shear region where convection is enhanced by the environmental vertical wind shear. Large horizontal gradients of wind speed are found in association with strong updrafts, and from the perturbation wind structure, the authors identify distinct vortices. The authors also argue that the tornado-scale vortices are related to horizontal roll vortices, and they suggest that the vortices may be related to the local large vertical wind shear that is present in the low-level eyewall.

Overall, this is an interesting study that contributes to our knowledge of intense small-scale vortices that are believed to be prevalent within the low-level eyewall of intense tropical cyclones. I have a number of minor scientific comments that are mostly related to requests for clarifications, but also include some areas where I'm not quite convinced that the authors' analysis demonstrates what is claimed. I also have a few more significant concerns. First, I think it is possible that the use of a moving average to define the reference state for wind speed may result in an exaggeration of the gradients in the perturbation winds, and that the azimuthal mean (or azimuthal-mean + low-wavenumber flow) may be a better choice for this analysis. Second, a study (Stern and Bryan 2018) has recently been published, that also used LES to examine these eyewall vortices, and so I think (in revision) that this current study should include some discussion of how their results may relate to those of Stern and Bryan (though I recognize that the authors submitted their manuscript just prior to the appearance online of the earlier study, so I don't mean this as a critique of this manuscript). Finally, it seems that a major result of this study is the finding that the eyewall vortices are related to pre-existing horizontal roll vortices within the boundary layer at and outside of the eyewall. I'm not fully convinced this is the case (though it may be), as the authors haven't really objectively defined the horizontal roll vortices that they see, and the existence of an updraft/downdraft couplet in the tornado-scale vortex isn't itself (in my view) necessarily a horizontal roll vortex (also see minor comments #25-26). Following revisions, I think that this study can be a nice contribution to the literature.

**Specific Major Comments**:

A.  Use of a moving average to define the reference state

    I think it may be problematic to use the 8-km moving average for calculating perturbation winds. This choice results in the much weaker tangential winds within the eye influencing the perturbation winds in the tornado-scale vortices, and vice versa. For example, in Fig. 3b, the perturbation flow within the eye is apparently anticyclonic, as the mean winds are much stronger than the local flow (because they include a region of the eyewall). This results in an

exaggerated characterization of the vortices, because the mean radial gradients are influencing the perturbation structure. I think a better choice would be to use the azimuthal mean (at a given radius) to define the perturbation winds. I see that the authors have examined something similar to this and they stated that they found similar results to their choice of the moving average. Still, I think the azimuthal mean is a more appropriate choice than the moving average.

B. Discussion of other recent related studies

With respect to observations of extreme local wind speeds and updrafts that are believed to be related to small-scale vortices, I think it would be worthwhile to discuss the recent study of Stern et al. (2016), who examined extreme updrafts and wind speeds observed by dropsondes. Also, Stern and Bryan (2018) very recently published a study using LES to investigate similar features as to what the authors examine in this manuscript. This was probably not available to the authors at the time that they submitted their manuscript, but given the similarity in some of the goals and methods of these studies, I think it would be worthwhile for the authors to add some discussion of how their results may relate to those of Stern and Bryan (2018). https://journals.ametsoc.org/doi/abs/10.1175/MWR-D-18-0041.1

C. Results that are not shown
There are a fairly large number of results that the authors refer to that aren't actually shown (given in minor comments below). This can be ok, but they need to make clear when a claim isn't explicitly shown by a figure. Also, these results that aren't shown probably shouldn't be included in the abstract (e.g., that the in nearly all vortices, there is also a broad downdraft).

**Specific Minor Comments**:

1. P5 l117
   "may be responsible for TC intensification" is too strong, I suggest changing to "may contribute to". It also might be a good idea to note here (I see it is mentioned later) that other subsequent studies (such as Bryan and Rotunno 2009) have found that this mechanism is unimportant for intensification.

2. P6 l126-131
   It's a bit unclear why a real case is chosen, but without any evaluation of the simulation in comparison to the observed storm. I'm guessing that the goal here is to examine a realistically sheared storm (and this may be easier to do in the real-case framework), but not to reproduce the evolution or structure of a specific real typhoon. This is ok, but the reasoning here should be made clearer. I note that Typhoon Matsa was not particularly intense, only an estimated 90 kt peak intensity, whereas it appears that the simulated storm here is stronger.

3. P6 l159-160

The authors state that they output 3-s data for a 22-min period at t=29 h.  This output is only used briefly on page 11 (with no figures shown), and so I think it would be better to move the description here to be part of the discussion of where it is used on p11.

4. P8 l172-173
Note that the persistence of the open eyewall isn't shown.

5. P8 l178
The shear given here is 5.2 m/s, but the figure caption gives 7.0 m/s.

6. P8 l181
The period here is stated to be 11 hours, but above it is given as 10 h.  Also, not that the RMW range is not shown.

7. P8 l187
Clarify that you are referring to the TC-scale shear-induced convective asymmetry here, not local enhancement of reflectivity around the tornado-scale vortices.

8. P9 l202 and Fig. 3b
The figure is somewhat confusing, because the labels for the identified vortices are not found where the actual features are.  It would be good to add a dot/circle (or some other symbol) to indicate the specific locations.

Also, I think it would be better to describe the convention for numbering the vortices here (where they are first introduced), rather than later when referring to Table 1.

9. P9 l209-211, p10 l213-221
The authors refer to the features examined here as "tornado-scale" vortices, and they discuss this in the context of the studies of Aberson et al. (2006), Aberson et al. (2017), and Marks et al. (2008).  But these prior studies did not refer to the features as "tornado-scale", and so this could be somewhat misleading.  The more recent study of Wurman and Kosiba (2018) did use this terminology, and I think it can be an appropriate choice.  But the authors should clarify how previous studies viewed these features.

10. P10 l217
Why is there a height threshold used in the definition here?

11. P10 l218
These thresholds are somewhat arbitrary.  Also, we don't really know that these are vortices simply from the vorticity threshold, as a region of high vorticity isn't necessarily a vortex.

12. P10 l231-232

I agree that these vortices may be responsible for the strongest wind gusts in TCs, and this is also consistent with the results shown in Stern and Bryan (2018).

13. P10 l227-238

I'm of the belief that the small-scale vortices probably don't have a substantial effect on the overall intensity evolution. However, I don't think the analyses in this study can really answer this question one way or the other, since we don't know how this simulation would have evolved in the absence of these features.

14. P11 l245-246

I note that this relationship between the vertical wind shear orientation and the spatial distribution of extreme updrafts is consistent with what Stern et al. (2016) found from dropsonde observations in many storms.

15. P11 l246-251

It could be good to discuss Stern and Bryan (2018) and Wurman and Kosiba (2018) here, as these studies both estimated the period for which these vortices/updrafts could be tracked (and with similar time periods as the authors have found).

16. P11 l254-255

I think it would be good to make clear that the authors are not directly identifying "vortices", but rather are identifying strong updrafts that are collocated with strong vorticity, and they are inferring that these are likely vortices (aside from the specific example vortices that they more directly identify).

17. P11 l257-258

That the updrafts/vortices are often found inward of the mean RMW is somewhat consistent with Stern and Bryan (2018), though in their study, they found that the strongest updrafts tended to occur more nearly at the RMW.

18. P11 l258

Clarify that you are referring to the w=20 m/s threshold, not the w=15 m/s threshold that has also been examined in this study.

19. P12 l260-261

Note that this result is not shown.

20. P12 l271

Please clarify what is meant by "using the smoothed fields". In what manner is the smoothing done?

21. P12 l271

Please define the Richardson number.

22. P12 l274

It would be helpful for the authors to more clearly explain why they are examining the Richardson number, what they expect it to tell them, and why they believe it should be related to the existence of these vortices. I see that the authors do so somewhat at the end of this paragraph, but I think it would be good to include this reasoning here where they introduce their analysis.

23. P13 l293-294

I'm not sure what this sentence ("The altitudes of the maximum vertical motions generally increase when the inflow layer deepens outward") means. Perhaps the authors are saying that the height of the features tends to be greater when they are found at larger radii?

24. Section 6

It's a bit confusing to have the vortices split into categories without first defining what the categories are. I think the distinctions should be brought up at the beginning of the section, along with information on how specifically the vortices are assigned to a category (is it subjective?), and a discussion of the physical reasoning for these classifications.

25. P14 l312-313

In my opinion, it is hard to tell if the features discussed here are indeed "closely associated with horizontal rolls". Is there any more objective evidence the authors can provide that can better demonstrate this claim?

26. P14 l318

It seems that the authors are concluding that these are roll vortices because there is a updraft/downdraft couplet, and so this implies a transverse circulation and horizontal vorticity. But I don't think this is really the same thing as what is traditionally referred to as a horizontal roll vortex, which generally have an elongated quasi-linear region of weak updrafts/downdrafts. These eyewall vortices will naturally have local updraft/downdraft couplets and large horizontal vorticity, but I don't think this makes them *necessarily* related to horizontal roll vortices.

27. P14 l323-324

In my view, we don't actually know whether the high thetae layer is an indication of transport from the eye. There is high thetae further outward as well in this cross section, so the elevated layer of high thetae does not have to have originated within the eye, although it may have. I think a trajectory analysis is necessary to have confidence on the origin of this air mass.

28. P15 l329-331

It isn't clear to me why/how the downward motion at 500 m is responsible for the high thetae layer. It also is unclear to me why the low thetae layer near the surface should have lower

thetae because it is in inflow. It's true that the mean radial gradient tends to be negative (and so mean radial advection tends to yield a negative tendency), but this is generally outweighed by other tendencies (such as surface fluxes).

29. P15 l332-333
    Again, I don't think we can know from the analysis here that the high thetae eye air is entrained into the eyewall.

30. P15 l342-343
    It looks to me that "~65 m/s" should be "~60 m/s", and that "~90 m/s" should be "~95 m/s".

31. P15 l345
    It isn't clear to me why the downward motion is "consistent" with the "strong wind speed jumps". What is the relationship here? I'm guessing that the authors are implying that strong wind gusts could be caused by vertical advection of higher momentum from above.

32. P16 l360-362
    Is the structure described here for this particular feature believed to be generally true for other such features? It's unclear if there is a robust signature here, as the authors are showing a single example.

33. P17 l377-378
    I think it is important to acknowledge here that this definition is somewhat arbitrary, and to reiterate that the frequency of these inferred vortices is very sensitive to the thresholds of this definition.

34. P17 l387
    Here, the authors refer to updrafts stronger than 15 m/s, but their definition given above is for 20 m/s.

35. P17 l390
    That the updraft is generally associated with a downdraft is not shown.

36. Fig. 1
    Make the caption clearer by changing "instantaneous and azimuthal maximum" to "maximum instantaneous and azimuthal-mean".

37. Fig. 2
    The red dots in 2a aren't defined in the caption. Change "solid circles" to "black circles". Give the height at which the RMW is evaluated here. Insert ", respectively" after "and the radius of maximum wind". Remove "(27h)".

38. Fig. 6

The "vertical slice" doesn't look exactly vertical to me.  Please clarify if it is "nearly" vertical".

39. Fig. 7

Please clarify if this cross section is purely in the radial dimension (as opposed to projecting onto the azimuthal dimension as well).

40. Fig. 9

Please clarify if the vertical velocity shown here is also a perturbation quantity.

**Technical Corrections**:

1. P2 l40

"the favorable location" is vague and somewhat confusing.  I suggest "the location along the inner edge of the eyewall", or something similar.

2. P3 l66

"by now" should be "for now".

3. P5 l109

"$f$" should be italicized.

4. P6 l134 and elsewhere

When expressing the lengths of a grid, the units should be "km", not "km$^2$".

5. P6 l134

Insert "grid" before "spacing".

6. P6 l136

"dimentional" should be "dimensional".

7. P6 l142

Insert "for" after "except".

8. P7 l145

Insert "of" after "temperature".

9. P7 l160

Use "29 h" instead of "the 30$^{th}$ hour".

10. P7 l162

"northern north west" should be "north northwest".

11. P7 l163-166

"instantaneous and azimuthal maximum" should be "maximum instantaneous and azimuthal-mean", with "maximum" presumably applying to both metrics. The following sentence "The instantaneous output…" can be removed, as it is redundant.

12. P8 l167, l169
    I think it would be better to put the smaller number first for these ranges.

13. P8 l176
    Insert "is" after "shear".

14. P8 l183
    "France" should be "Frances".

15. P9 l194
    "filed" should be "field.

16. P9 l198
    "wind" should be "window".

17. P9 l209
    Aberson et al. (2016) should be Aberson et al. (2017).

18. P10 l233
    Insert "some" before "previous".

19. P10 l235
    "Montegomery" should be "Montgomery".

20. P10 l235
    "azimuthal" should be "azimuthal-mean"

21. P12 l272
    Insert "vortex" after "scale".

22. P12 l274-275
    Stern et al. (2016) is cited here, but it doesn't appear in the list of references.

23. P14 l305
    Move "inward" from end of sentence to right after "kilometers".

24. P14 l318
    "frank" is a typo here. I think that the authors mean "flank". Also, "rolling" should be "roll" (also where it is used elsewhere).

25. P15 l327
    Replace "To the right" with "Outward".

26. P15 l328

Change "the category" to "this category", and insert "of vortices" after "category".

27. P15 l328
It's ambiguous which feature "the lower-altitude high thetae layer" refers to.  Please clarify.

28. P15 l329
Change "does not" to "is not".

29. P15 l337
I think that the authors may mean Fig. 8a and not Fig. 8b.  Also, I think they may mean Fig. 3b and not Fig. 2b.

30. P16 l359
"Figures 10a" should be "Fig. 10a".

31. P16 l367
Insert "with" after "associated".  Insert "the" before "complicated".

32. P17 l371
Change "nesting" to "nested".

33. P17 l382
There is an extra period after "layer".

34. Fig. 4
"500-km" should be "500-m".

35. Fig. 8
Fig. 8b has "400 m", but the caption says "500 m".

36. Fig. 10
The last sentence of the caption is a duplicate.

---

## Referee Comment (RC4) · Anonymous Referee #2 · 1 Nov 2018

This paper describes the characteristics of relatively intense tornado-scale vortices in a high-resolution numerical simulation of a mature tropical cyclone under environmental conditions resembling those of Typhoon Matsa (2005). It is found that the simulated vortices have locations and basic properties that are broadly consistent with limited observations. An effort is made to classify the vortices into 3 distinct categories. In my view, the article is well organized and provides useful information that is adequately summarized in the abstract and section 7. Moreover, I did not catch any obvious mistakes of major consequence. On the other hand, I was somewhat disappointed not to see a rigorous analysis of the generation and decay of a tornado-scale vortex belonging to any of the 3 categories. High-resolution TC simulations showing tornado-scale vor-

tices are not unprecedented [e.g., Stern and Bryan 2014], and it seems to me that the most interesting scientific questions pertain to the formation and decay mechanisms.

Below are some minor comments that might be worth considering before official publication.

Minor Comments:

1. The paper cites an earlier study suggesting that grid-spacing less than 100 m is necessary for simulating the development of tornado-scale vortices. However, it is not entirely clear to me that simulating the 1-2 km structures of interest requires 37-m horizontal grid spacing, especially since the vertical grid spacing is (apparently) of order 100 m in the boundary layer. A brief comment on what happens to the tornado-scale vortices when the finest horizontal grid is removed in the present numerical experiment might be worthwhile.

2. There is a recent LES study by Ito et al. [Scientific Reports, 7.1, 3798 (2017)] that addresses the variation of roll structure with location in a TC boundary layer. Perhaps the authors should to try to connect the aforementioned study to theirs.

3. Since this article pertains to coherent structures having large horizontal components of relative vorticity, it might be a good idea to specify upfront that the term "relative vorticity" in this paper (presumably) refers to the vertical relative vorticity.

4. Lines 170-171: In my view, it seems a little awkward to introduce tornado-scale vortices as small-scale features that are distinct from horizontal rolls, but later show that they incorporate horizontal rolls (in some sense). That said, I am not sure that any changes need to be made in response to the preceding comment.

5. Lines 256-258: This statement (added after the first review) needs to be rewritten. To begin with, the statement fails to clarify whether the azimuthally averaged wind speed is an azimuthal average of the total horizontal wind speed or of the tangential (azimuthal) velocity. Of lesser importance, "are directly" should be "are obtained directly", and

there should probably be a comma after "time-averaging".

6. Line 374: To facilitate quantitative comparison with future studies, I think that it might be worthwhile to more precisely define the Richardson number (with an equation).

7. Lines 426-428: The wording suggests (to me) that the cited studies definitively showed that the wind speed bands are connected to vertical momentum transport by the rolls, but such an interpretation is challenged by the final sentence of the paragraph. I would consider revising the paragraph so as not to mislead the reader upfront.

8. Line 130: I suggest changing "the similar features as revealed with the limited observational data" to "features similar to those revealed with limited observational data".

9. Lines 156-159: This section of the paragraph tries to say too much in one sentence.

10. Line 275: I believe that "France" should be "Frances".

11. Line 290: "wind size" should be "window size".

12. Line 325: I would change "the mesovortices" to "mesovortices".

13. Line 339: I might remove "consecutive" or change it to "continuous".

14. Lone 350: "close to RMW" should be "close to the RMW".

15. Line 364: "tornado-scale" should be "tornado-scale vortex".

16. Line 404: I would change "Besides" to "In addition".

17. Line 410: Should "frank" be "flank"?

18. Line 425: I might change "entrained" to "locally entrained".

19. Line 459: "associated strong turbulence" should be "associated with strong turbulence".

20. Lines 463-465: "nesting grids" should be "nested grids"; "shows the similar features

as revealed with the limited observations" should be something like "shows features similar to those revealed with limited observations"; "favorable" should probably be "favored".

21. Lines 478-480: These sentences seem largely redundant with the preceding paragraph.
* * *

---

## Referee Comment (RC5) · Anonymous Referee #1 · 2 Nov 2018

This manuscript documents the small-scale vortices in the tropical cyclone boundary layer found in a two-way nested WRF-LES set up using the large-scale conditions of a real typhoon. The results are interesting and the presentation is quite clear. I have only a few minor comments about the model setup and interpretation of results.

Minor Comments:

1. The 100-m vertical resolution is relatively coarse for adequately resolving structures like the high theta_e layers shown in Fig. 7b. It is also coarse compared to the 37-m horizontal resolution. A higher vertical resolution is also desirable for capturing the strength and scale of the horizontal roll vortices mentioned by the authors (Fig. 3a).

[Figure]

Have the authors done any sensitivity tests to examine the impact of vertical resolution on the structure and distribution of the small-scale vortices focused on in the work?

2. I couldn't quite infer the exact connection between the "quasi-linear bands" and what is shown in Fig. 8 (paragraph starting on Line 334). Are the authors implying that the wind speed horizontal variablity associated with the quasi-linear features could explain in part the wind speed jump associated with the "tornado-scale vortices"? Please be more explicit.

3. By categorizing the vortices into 3 groups, are the authors suggesting that they are generated/maintained by different physical mechanisms? Could they simply represent different phases in the life cycle of these coherent structures?

---

## Author Comment (AC1) · 24 Dec 2018

The comment was uploaded in the form of a supplement: https://www.atmos-chem-phys-discuss.net/acp-2018-787/acp-2018-787-AC1-supplement.zip

---

## Author Comment (AC2) · 24 Dec 2018

The comment was uploaded in the form of a supplement:
https://www.atmos-chem-phys-discuss.net/acp-2018-787/acp-2018-787-AC2-supplement.zip

---

## Author Comment (AC3) · 24 Dec 2018

The comment was uploaded in the form of a supplement:
https://www.atmos-chem-phys-discuss.net/acp-2018-787/acp-2018-787-AC3-supplement.zip

---

## Author Comment (AC4) · 24 Dec 2018

The comment was uploaded in the form of a supplement:
https://www.atmos-chem-phys-discuss.net/acp-2018-787/acp-2018-787-AC4-
supplement.zip

---

## Author Comment (AC5) · 24 Dec 2018

The comment was uploaded in the form of a supplement:
https://www.atmos-chem-phys-discuss.net/acp-2018-787/acp-2018-787-AC5-supplement.zip

---

## Author Response (AR1)

**Reply to Referee #3**

Thank you very much for your valuable comments. In the future we will examine the sustained wind speed by using different criteria to figure out which averaging time is more adequate to represent maximum sustained wind in a TC.

The description of grid resolution and time step in our experiment are listed as following:

| Domain | Grid resolution | Time step |
|--------|-----------------|-----------|
| D1 | 27km | 30.00s |
| D2 | 9km | 10.00s |
| D3 | 3km | 3.33s |
| D4 | 1km | 1.11s |
| D5 | 333.3m | 0.37s |
| D6 | 111.1m | 0.12s |
| D7 | 37.03m | 0.04s |

Minor corrections:

*1. P5, line 108, "When the horizontal resolution was decreased… should read "When the horizontal resolution was increased…"*

*2. P9, line 198, "…wind size…" should be "…window size…"*

We have revised the manuscript and the above errors have been corrected.

**Reply to Referee #4**

*Some comments and suggestions are provided below:*

*Line 92-94: "Such strong turbulence was also observed in Hurricane Isabel (2003) and Felix (2007) at different altitudes (Aberson et al. 2006; Aberson et al .2007)". It is better to list the exact altitudes of this "different altitudes" to make sure these are related to TC BL turbulence.*

The extreme updraft (~25 m/s) and horizontal wind (107 m/s) was found at about 1.5 km in Hurricane Isabel (2003). The extreme updraft (~ 31 m/s) was found at about 3km in Hurricane Felix (2007). These extreme updrafts are consistent with the analysis by Stern et al. (2016).

The sentence has rewritten as: Such strong turbulence was also observed in Hurricanes Isabel (2003) below 3-km (Aberson et al. 2006; Aberson et al. 2017).

*2. Line 94-96: "Understanding of the structure and evolution of the ... severe turbulence." This sentence doesn't match the logic. The reason to understanding of this small structure turbulence should be it is important for determining storm intensity, it should not be hard to observe. Using numerical simulation is because it is hard to observe.*

The sentence has been revised.

*3. Line 132-145: The finest resolution of horizontal resolution of this simulation is 37 meters, while the vertical resolution is only 75 levels. This concerns as the ratio of horizontal resolution and the vertical resolution could play a big role in the 3D simulations.*

We understand your concern. The vertical resolution in the innermost domain is relatively coarse compared to the horizontal spacing of 37 m.  We did not run experiments to examine the sensitivity to the vertical resolution because of the limit of the computation resource. In fact, we attempted to increase the vertical resolution, but the model cannot run on Tianhe-2 computer. For this reason, we conducted the LES-111 experiment (111.1m horizontal resolution) with 12 vertical levels below 1km. In LES-111 experiment, the vertical resolution and horizontal resolution are comparable in the TC boundary layer. The near-surface linear coherent structures and tornado-scale vortex (TSV) simulated in LES-111 are similar to those in the LES-37 experiment. In the revised manuscript, we have added a brief description about the issue.

*4. Line 156-157: "we will focus on the hourly output from 26h to 36h." Since this is tornado scale feature and the horizontal resolution reaches 37m, hourly output is too coarse and would miss some features. Suggest taking a more aggressive evaluation of output of the order of minutes (at least 15 minutes).*

You are right. The hourly output is to coarse to analyze the tornado scale features. For this reason, we stored the 3-second model output to examine the evolution of the simulated TSVs. Since the 3-second output does not contain the thermodynamic variables, we need rerun the experiment for further analysis.

*5. It is better to indicate the red dots as tornado-scale vortices in Fig.2a in figure caption.*

The figure caption has been rewritten.

**Reply to Referee #5**

**General Comments**:

In this study, the authors use WRF-LES to simulate a quasi-idealized tropical cyclone (in the environment of a real TC), for the purpose of investigating tornado-scale vortices. They find such vortices along the inner eyewall, concentrated in the left-of-shear region where convection is enhanced by the environmental vertical wind shear. Large horizontal gradients of wind speed are found in association with strong updrafts, and from the perturbation wind structure, the authors identify distinct vortices. The authors also argue that the tornado-scale vortices are related to horizontal roll vortices, and they suggest that the vortices may be related to the local large vertical wind shear that is present in the low-level eyewall.

Overall, this is an interesting study that contributes to our knowledge of intense small-scale vortices that are believed to be prevalent within the low-level eyewall of intense tropical cyclones. I have a number of minor scientific comments that are mostly related to requests for clarifications, but also include some areas where I'm not quite convinced that the authors' analysis demonstrates what is claimed. I also have a few more significant concerns. First, I think it is possible that the use of a moving average to define the reference state for wind speed may result in an exaggeration of the gradients in the perturbation winds, and that the azimuthal mean (or azimuthal-mean + low-wavenumber flow) may be a better choice for this analysis. Second, a study (Stern and Bryan 2018) has recently been published, that also used LES to examine these eyewall vortices, and so I think (in revision) that this current study should include some discussion of how their results may relate to those of Stern and Bryan (though I recognize that the authors submitted their manuscript just prior to the appearance online of the earlier study, so I don't mean this as a critique of this manuscript). Finally, it seems that a major result of this study is the finding that the eyewall vortices are related to pre-existing horizontal roll vortices within the boundary layer at and outside of the eyewall. I'm not fully convinced this is the case (though it may be), as the authors haven't really objectively defined the horizontal roll vortices that they see, and the existence of an updraft/downdraft couplet in the tornado-scale vortex isn't itself (in my view) necessarily a horizontal roll vortex (also see minor comments #25-26). Following revisions, I think that this study can be a nice contribution to the literature.

Specific Major Comments:

A. *Use of a moving average to define the reference state*

   *I think it may be problematic to use the 8-km moving average for calculating perturbation winds. This choice results in the much weaker tangential winds*

*within the eye influencing the perturbation winds in the tornado-scale vortices, and vice versa. For example, in Fig. 3b, the perturbation flow within the eye is apparently anticyclonic, as the mean winds are much stronger than the local flow (because they include a region of the eyewall). This results in an exaggerated characterization of the vortices, because the mean radial gradients are influencing the perturbation structure. I think a better choice would be to use the azimuthal mean (at a given radius) to define the perturbation winds. I see that the authors have examined something similar to this and they stated that they found similar results to their choice of the moving average. Still, I think the azimuthal mean is a more appropriate choice than the moving average.*

Thank you for your suggestion. Based on the numerical study conducted by Green et al. (2015), we chose the 8-km moving average for calculating perturbation winds. We have checked the results of 2 different methods for calculation perturbation winds. One is the 8-km moving average filter method, and the other is low-wavenumber (azimuthal-mean + wavenumber1-3) flow filer method. From the attached figure (Fig. A2), we can see an exaggerated characterization of the vortices indeed exist inside the eyewall by using the 8-km moving average, but the two different methods have little effect on the perturbation wind field associated with the tornado-scale vortex. We have mentioned this in the revised manuscript.

B. *Discussion of other recent related studies*
   *With respect to observations of extreme local wind speeds and updrafts that are believed to be related to small-scale vortices, I think it would be worthwhile to discuss the recent study of Stern et al. (2016), who examined extreme updrafts and wind speeds observed by dropsondes. Also, Stern and Bryan (2018) very recently published a study using LES to investigate similar features as to what the authors examine in this manuscript. This was probably not available to the authors at the time that they submitted their manuscript, but given the similarity in some of the goals and methods of these studies, I think it would be worthwhile for the authors to add some discussion of how their results may relate to those of Stern and Bryan (2018).*

Thank you for providing the latest references. We added the discussion on the study of Stern et al. (2016) and Stern and Bryan (2018) in the revised manuscript.

C. *Results that are not shown*
   *There are a fairly large number of results that the authors refer to that aren't actually shown (given in minor comments below). This can be ok, but they need to make clear when a claim isn't explicitly shown by a figure. Also, these*

*results that aren't shown probably shouldn't be included in the abstract (e.g., that the in nearly all vortices, there is also a broad downdraft).*

In the revised manuscript, we have explicitly indicated the figures that are not shown.

[Figure]

Figure A2 Comparison of the perturbation wind field as shown in Fig. 3b from (a) the result of low-wavenumber (azimuthal-mean + wavenumber 1-3) flow filer method and (b) the result of the 8-km moving average filter.

Specific Minor Comments:

1. *P5 l117*

   *"may be responsible for TC intensification" is too strong, I suggest changing to "may contribute to". It also might be a good idea to note here (I see it is mentioned later) that other subsequent studies (such as Bryan and Rotunno 2009) have found that this mechanism is unimportant for intensification.*

   Changed.

2. P6 l126-131

   *It's a bit unclear why a real case is chosen, but without any evaluation of the simulation in comparison to the observed storm. I'm guessing that the goal here is to examine a realistically sheared storm (and this may be easier to do in the real-case framework), but not to reproduce the evolution or structure of a specific real typhoon. This is ok, but the reasoning here should be made clearer. I note that Typhoon Matsa was not particularly intense, only an estimated 90 kt peak intensity, whereas it appears that the simulated storm here is stronger.*

   We used the real case because we want to make the simulated TC evolves in a realistic environment. The typhoon was first simulated in Wu and Chen (2016) without using the LES. For convenience, we used the initial and boundary conditions in this study. As you mentioned, we found that the occurrence of the simulated tornado-scale vortices is closely associated with the environmental shear.

   You are right. The estimated intensity of Typhoon Matsa was not very intense. It is interesting to note that its intensity is close to the azimuthal mean maximum wind speed of the simulated TC although the simulated gust winds are much stronger. It is possible that the simulated TC intensity is stronger than the real typhoon. It is also possible that the maximum sustained wind speed was missed in the observation.

3. *P6 l159-160*

   *The authors state that they output 3-s data for a 22-min period at t=29 h. This output is only used briefly on page 11 (with no figures shown), and so I think it would be better to move the description here to be part of the discussion of where it is used on p11.*

   Done.

4. *P8 l172-173*

   *Note that the persistence of the open eyewall isn't shown.*

We only show the simulated radar reflectivity in Fig. 2. In the revised statement, we explicitly mention the other figures are not shown.

5. *P8 l178*

   *The shear given here is 5.2 m/s, but the figure caption gives 7.0 m/s.*

   The shear is 7.0 m/s. Corrected.

6. *P8 l181*

   *The period here is stated to be 11 hours, but above it is given as 10 h.    Also, not that the RMW range is not shown.*

   Corrected.

7. *P8 l187*

   *Clarify that you are referring to the TC-scale shear-induced convective asymmetry here, not local enhancement of reflectivity around the tornado-scale vortices.*

   Clarified.

8. *P9 l202 and Fig. 3b*

   *The figure is somewhat confusing, because the labels for the identified vortices are not found where the actual features are.   It would be good to add a dot/circle (or some other symbol) to indicate the specific locations. Also, I think it would be better to describe the convention for numbering the vortices here (where they are first introduced), rather than later when referring to Table 1.*

   The figure is revised.

9. *P9 l209-211, p10 l213-221*

   *The authors refer to the features examined here as "tornado-scale" vortices, and they discuss this in the context of the studies of Aberson et al. (2006), Aberson et al. (2017), and Marks et al. (2008).    But these prior studies did not refer to the features as "tornado-scale", and so this could be somewhat misleading.    The more recent study of Wurman and Kosiba (2018) did use this terminology, and I think it can be an appropriate choice.    But the authors should clarify how previous studies viewed these features.*

   This has been clarified in the revised manuscript.

10. *P10 l217*

    *Why is there a height threshold used in the definition here?*

    In this study, we limit our discussion in the TC boundary.

11. *P10 l218*

   *These thresholds are somewhat arbitrary. Also, we don't really know that these are vortices simply from the vorticity threshold, as a region of high vorticity isn't necessarily a vortex.*

   These thresholds are based on the observation. We examined all of the identified TSVs and found that all of the TSVs are associated with strong horizontal circulation.

12. *P10 l231-232*

   *I agree that these vortices may be responsible for the strongest wind gusts in TCs, and this is also consistent with the results shown in Stern and Bryan (2018).* The reference is cited in the revised manuscript.

13. *P10 l227-238*

   *I'm of the belief that the small-scale vortices probably don't have a substantial effect on the overall intensity evolution. However, I don't think the analyses in this study can really answer this question one way or the other, since we don't know how this simulation would have evolved in the absence of these features.*

   You are right. Our statement is based on the occurrence of the simulated TSVs. As shown in Figure 1, the azimuthal-mean maximum wind speed does not show any jump at 27 h, when there are 10 identified tornado-scale vortices.

14. *P11 l245-246*

   *I note that this relationship between the vertical wind shear orientation and the spatial distribution of extreme updrafts is consistent with what Stern et al. (2016) found from dropsonde observations in many storms.*

   The reference has been cited in the revised manuscript.

15. *P11 l246-251*

   *It could be good to discuss Stern and Bryan (2018) and Wurman and Kosiba (2018) here, as these studies both estimated the period for which these vortices/updrafts could be tracked (and with similar time periods as the authors have found).*

   Thank you for your suggestion. We have revised the description.

16. *P11 l254-255*

*I think it would be good to make clear that the authors are not directly identifying "vortices", but rather are identifying strong updrafts that are collocated with strong vorticity, and they are inferring that these are likely vortices (aside from the specific example vortices that they more directly identify).*

We have examined all of the 24 TSVs and all of them are associated with strong horizontal circulation.

17. *P11 l257-258*

    *That the updrafts/vortices are often found inward of the mean RMW is somewhat consistent with Stern and Bryan (2018), though in their study, they found that the strongest updrafts tended to occur more nearly at the RMW.*

    We have cited this paper in the revised manuscript.

18. *P11 l258*

    *Clarify that you are referring to the w=20 m/s threshold, not the w=15 m/s threshold that has also been examined in this study.*

    Clarified.

19. *P12 l260-261*

    *Note that this result is not shown.*

    Specified.

20. *P12 l271*

    *Please clarify what is meant by "using the smoothed fields". In what manner is the smoothing done?*

    We chose the 8-km moving average for calculating perturbation winds. It has been clarified in the revised manuscript.

21. *P12 l271*

    *Please define the Richardson number.*

    Added.

22. *P12 l274*

    *It would be helpful for the authors to more clearly explain why they are examining the Richardson number, what they expect it to tell them, and why they believe it should be related to the existence of these vortices. I see that the authors do so somewhat at the end of this paragraph, but I think it would be good to include this reasoning here where they introduce their analysis.*

    We have included the expression and more information about Ri in this revision.

23. *P13 l293-294*

*I'm not sure what this sentence ("The altitudes of the maximum vertical motions generally increase when the inflow layer deepens outward") means. Perhaps the authors are saying that the height of the features tends to be greater when they are found at larger radii?*

You are right. The sentence has been revised.

24. Section 6

*It's a bit confusing to have the vortices split into categories without first defining what the categories are. I think the distinctions should be brought up at the beginning of the section, along with information on how specifically the vortices are assigned to a category (is it subjective?), and a discussion of the physical reasoning for these classifications.*

You are right. In this study, we subjectively split the vortices into three categories based on its vertical structure, especially in terms of its vertical extension, stratification and near-surface wind jump.

25. *P14 l312-313*

*In my opinion, it is hard to tell if the features discussed here are indeed "closely associated with horizontal rolls". Is there any more objective evidence the authors can provide that can better demonstrate this claim?*

The horizontal roll is indicated in the new Fig. 6b and Fig. 6c.

26. *P14 l318*

*It seems that the authors are concluding that these are roll vortices because there is a updraft/downdraft couplet, and so this implies a transverse circulation and horizontal vorticity. But I don't think this is really the same thing as what is traditionally referred to as a horizontal roll vortex, which generally have an elongated quasi-linear region of weak updrafts/downdrafts. These eyewall vortices will naturally have local updraft/downdraft couplets and large horizontal vorticity, but I don't think this makes them necessarily related to horizontal roll vortices.*

Previous studies suggest that the near surface quasi-linear coherent structures are associated with horizontal roll vortices. A typical tornado-scale vortex contains an updraft/downdraft couplet and the updraft in tornado-scale vortex are stronger than in horizontal roll vortices.

27. *P14 l323-324*

*In my view, we don't actually know whether the high thetae layer is an indication of transport from the eye. There is high thetae further outward as well in this cross section, so the elevated layer of high thetae does not have to have originated within the eye, although it may have. I think a trajectory analysis is necessary to have confidence on the origin of this air mass.*

The sentence has been removed.

28. *P15 l329-331*

*It isn't clear to me why/how the downward motion at 500 m is responsible for the high thetae layer. It also is unclear to me why the low thetae layer near the surface should have lower thetae because it is in inflow. It's true that the mean radial gradient tends to be negative (and so mean radial advection tends to yield a negative tendency), but this is generally outweighed by other tendencies (such as surface fluxes).*

The sentence has been removed.

29. *P15 l332-333*

*Again, I don't think we can know from the analysis here that the high thetae eye air is entrained into the eyewall.*

You are right. The related sentence has been revised.

30. *P15 l342-343*

*It looks to me that "~65 m/s" should be "~60 m/s", and that "~90 m/s" should be "~95 m/s".*

Corrected.

31. *P15 l345*

*It isn't clear to me why the downward motion is "consistent" with the "strong wind speed jumps". What is the relationship here? I'm guessing that the authors are implying that strong wind gusts could be caused by vertical advection of higher momentum from above.*

You are right.

32. *P16 l360-362*

*Is the structure described here for this particular feature believed to be generally true for other such features? It's unclear if there is a robust signature here, as the authors are showing a single example.*

The feature is common for tornado-scale vortices in the category. Tornado-scale vortices in the third category mainly occur in the statically stable stratification.

33. *P17 l377-378*

   *I think it is important to acknowledge here that this definition is somewhat arbitrary, and to reiterate that the frequency of these inferred vortices is very sensitive to the thresholds of this definition.*

   You are right. We have revised the sentence.

34. *P17 l387*

   *Here, the authors refer to updrafts stronger than 15 m/s, but their definition given above is for 20 m/s.*

   Corrected.

35. *P17 l390*

   *That the updraft is generally associated with a downdraft is not shown.*

   We explicitly mention figures that are not shown. The updraft and downdraft couplet can be seen in Fig. 6b in the revised manuscript.

36. *Fig. 1*

   *Make the caption clearer by changing "instantaneous and azimuthal maximum" to "maximum instantaneous and azimuthal-mean".*

   Corrected.

37. *Fig. 2*

   *The red dots in 2a aren't defined in the caption.   Change "solid circles" to "black circles".   Give the height at which the RMW is evaluated here.   Insert ", respectively" after "and the radius of maximum wind".   Remove "(27h)".*

   Corrected and added.

38. *Fig. 6*

   *The "vertical slice" doesn't look exactly vertical to me.   Please clarify if it is "nearly" vertical".*

   Corrected.

39. *Fig. 7*

   *Please clarify if this cross section is purely in the radial dimension (as opposed to projecting onto the azimuthal dimension as well).*

   This cross section is in the radial dimension. We have revised the description in our manuscript.

40. *Fig. 9*

   *Please clarify if the vertical velocity shown here is also a perturbation quantity.*

Yes. It is a perturbation quantity.

Technical Corrections:

1. *P2 140*

   *"the favorable location" is vague and somewhat confusing. I suggest "the location along the inner edge of the eyewall", or something similar.*

   Changed.

2. *P3 166*

   *"by now" should be "for now".*

   Corrected.

3. *P5 1109*

   *"f" should be italicized.*

   Corrected.

4. *P6 1134 and elsewhere*

   *When expressing the lengths of a grid, the units should be "km", not "km2".*

   Corrected.

5. *P6 1134*

   *Insert "grid" before "spacing".*

   Corrected.

6. *P6 1136*

   *"dimentional" should be "dimensional".*

   Corrected.

7. *P6 1142*

   *Insert "for" after "except".*

   Added.

8. *P7 1145*

   *Insert "of" after "temperature".*

   Added.

9. P7 1160

   *Use "29 h" instead of "the 30th hour".*

   Corrected.

10. *P7 1162*

    *"northern north west" should be "north northwest".*

Corrected.

11. *P7 l163-166*

   *"instantaneous and azimuthal maximum" should be "maximum instantaneous and azimuthal mean", with "maximum" presumably applying to both metrics. The following sentence "The instantaneous output…" can be removed, as it is redundant.*

   Corrected.

12. *P8 l167, l169*

    *I think it would be better to put the smaller number first for these ranges.*
    Corrected.

13. *P8 l176*

    *Insert "is" after "shear".*

    Added.

14. *P8 l183*

    *"France" should be "Frances".*

    Corrected.

15. *P9 l194*

    *"filed" should be "field.*

    Corrected.

16. *P9 l198*

    *"wind" should be "window".*

    Corrected.

17. *P9 l209*

    *Aberson et al. (2016) should be Aberson et al. (2017).*

    Corrected.

18. *P10 l233*

    *Insert "some" before "previous".*

    Added.

19. P10 l235

    *"Montegomery" should be "Montgomery".*

    Corrected.

20. *P10 l235*

    *"azimuthal" should be "azimuthal-mean"*

    Corrected.

21. *P12 l272*

    *Insert "vortex" after "scale".*

    Corrected.

22. *P12 l274-275*

    *Stern et al. (2016) is cited here, but it doesn't appear in the list of references.*

    Added.

23. P14 l305

    Move "inward" from end of sentence to right after "kilometers".

    Corrected.

24. *P14 l318*

    *"frank" is a typo here.   I think that the authors mean "flank".   Also, "rolling" should be "roll" (also where it is used elsewhere).*

    Corrected.

25. *P15 l327*

    *Replace "To the right" with "Outward".*

    Corrected.

26. *P15 l328*

    *Change "the category" to "this category", and insert "of vortices" after "category".*

    Corrected.

27. *P15 l328*

    *It's ambiguous which feature "the lower-altitude high thetae layer" refers to. Please clarify.*

    The sentence has been rewritten.

28. *P15 l329*

    *Change "does not" to "is not".*

    Corrected.

29. *P15 l337*

    *I think that the authors may mean Fig. 8a and not Fig. 8b.   Also, I think they may mean Fig. 3b and not Fig. 2b.*

    Corrected.

30. *P16 l359*

    *"Figures 10a" should be "Fig. 10a".*

    Corrected.

31. *P16 l367*

*Insert "with" after "associated".   Insert "the" before "complicated".*

Added.

32. *P17 1371*

    *Change "nesting" to "nested".*

    Corrected.

33. *P17 1382*

    *There is an extra period after "layer".*

    Corrected.

34. *Fig. 4*

    *"500-km" should be "500-m".*

    Corrected.

35. Fig. 8

    Fig. 8b has "400 m", but the caption says "500 m".

    Corrected.

36. Fig. 10

    The last sentence of the caption is a duplicate.

    Deleted.

**Reply to Referee #2**

*This paper describes the characteristics of relatively intense tornado-scale vortices in a high-resolution numerical simulation of a mature tropical cyclone under environmental conditions resembling those of Typhoon Matsa (2005). It is found that the simulated vortices have locations and basic properties that are broadly consistent with limited observations. An effort is made to classify the vortices into 3 distinct categories. In my view, the article is well organized and provides useful information that is adequately summarized in the abstract and section 7. Moreover, I did not catch any obvious mistakes of major consequence. On the other hand, I was somewhat disappointed not to see a rigorous analysis of the generation and decay of a tornado-scale vortex belonging to any of the 3 categories. High-resolution TC simulations showing tornado-scale vortices are not unprecedented [e.g., Stern and Bryan 2014], and it seems to me that the most interesting scientific questions pertain to the formation and decay mechanisms. Below are some minor comments that might be worth considering before official publication.*

We absolutely agree with you that this manuscript does not include a rigorous analysis of the generation and decay of the tornado-scale vortex. As we mentioned in the manuscript, the model output is regularly stored at 1-h intervals, and a few variables during a 22-min period from the 30th hour are also stored at 3-s intervals. In this study, we mainly used 1-hour outputs to check the structures of tornado-scale vortices. We think that considerable analysis is needed to understand the mechanisms for the generation and decay of the tornado-scale vortex. We plan to rerun the experiment by adding more variables in the 3-s output and investigate the mechanisms for the generation and decay of the tornado-scale vortex in the future. We have added some discussions in the revised manuscript.

*1. The paper cites an earlier study suggesting that grid-spacing less than 100 m is necessary for simulating the development of tornado-scale vortices. However, it is not entirely clear to me that simulating the 1-2 km structures of interest requires 37-m horizontal grid spacing, especially since the vertical grid spacing is(apparently)of order 100 m in the boundary layer. A brief comment on what happens to the tornado-scale vortices when the finest horizontal grid is removed in the present numerical experiment might be worthwhile.*

Our experiment contains 12 vertical levels below 1 km. We also conducted an experiment with the resolution of 111 m in the innermost domain. In the experiment, the vertical resolution and horizontal resolution are comparable in TC boundary layer. The tornado-scale vortex (TSV) mentioned in observations can also be found in the experiment. In the attached figure, we can see a simulated TSV in the experiment, similar to the TSV in Figure 6. The maximum vertical motion is 21.3 m s$^{-1}$ at 500 m and the maximum relative vertical vorticity is $0.11$ s$^{-1}$. In the revised manuscript, we have added a brief description about the issue.

[Figure]

Fig. A1 A simulated TSV in the LES-111 experiment. The description of the figure is the same as Fig. 6 in the manuscript, except for LES-111 experiment.

*2. There is a recent LES study by Ito et al. [Scientific Reports, 7.1, 3798 (2017)] that addresses the variation of roll structure with location in a TC boundary layer. Perhaps the authors should to try to connect the aforementioned study to theirs.*

Thank you for providing the reference. We have introduced the research conducted by Ito et al. (2017) in the Introduction.

*3. Since this article pertains to coherent structures having large horizontal components of relative vorticity, it might be a good idea to specify upfront that the term "relative vorticity" in this paper (presumably) refers to the vertical relative vorticity.*

This is a good idea. We have done this in the revised manuscript.

*4. Lines 170-171: In my view, it seems a little awkward to introduce tornado-scale vortices as small-scale features that are distinct from horizontal rolls, but later show that they incorporate horizontal rolls (in some sense). That said, I am not sure that any changes need to be made in response to the preceding comment.*

In this manuscript, we focus mostly the tornado-scale vortices and the calculated vorticity is the vertical component of relative vorticity. The simulated tornado-scale vortices are distinct from horizontal rolls because the strong updrafts are always accompanied by strong horizontal circulations.

*5. Lines 256-258: This statement (added after the first review) needs to be rewritten. To begin with, the statement fails to clarify whether the azimuthally averaged wind speed is an azimuthal average of the total horizontal wind speed or of the tangential (azimuthal) velocity. Of lesser importance, "are directly" should be "are obtained directly", and there should probably be a comma after "time-averaging".*

Thank you. The statement has been revised.

*6. Line 374: To facilitate quantitative comparison with future studies, I think that it might be worthwhile to more precisely define the Richardson number (with an equation).*

We have added some statement on Richardson number. The gradient Richardson number, Ri, has largely been used as a criterion for assessing the stability of stratified shear flow. It is defined by

$$R_i = \frac{N^2}{S^2} \qquad (1)$$

$N^2 = g\frac{\partial \ln \theta_e}{\partial z}$ is the square of Brunt–Väisälä frequency and $S^2 = (\frac{\partial u}{\partial z})^2 + (\frac{\partial v}{\partial z})^2$ is the square of vertical shear of the horizontal velocity, g is the gravity acceleration, $\theta_e$ is the equivalent potential temperature, u is the zonal wind speed and v is the meridional wind speed.

*7. Lines 426-428: The wording suggests (to me) that the cited studies definitively showed that the wind speed bands are connected to vertical momentum transport by the rolls, but such an interpretation is challenged by the final sentence of the paragraph. I would consider revising the paragraph so as not to mislead the reader upfront.*

The sentence has been revised.

*8. Line 130: I suggest changing "the similar features as revealed with the limited observational data" to "features similar to those revealed with limited observational data".*

The sentence has been rewritten as you suggest.

*9. Lines 156-159: This section of the paragraph tries to say too much in one sentence.*

The sentence has been rewritten as you suggest.

*10. Line 275: I believe that "France" should be "Frances".*

Corrected. Thank you!

*11. Line 290: "wind size" should be "window size".*

Corrected.

*12. Line 325: I would change "the mesovortices" to "mesovortices".*

Corrected.

*13. Line 339: I might remove "consecutive" or change it to "continuous".*

Corrected.

*14. Lone 350: "close to RMW" should be "close to the RMW".*

Corrected.

*15. Line 364: "tornado-scale" should be "tornado-scale vortex".*

Corrected.

*16. Line 404: I would change "Besides" to "In addition".*

Changed.

*17. Line 410: Should "frank" be "flank"?*

Corrected.

*18. Line 425: I might change "entrained" to "locally entrained".*

Changed.

*19. Line 459: "associated strong turbulence" should be "associated with strong turbulence".*

Corrected.

*20. Lines 463-465: "nesting grids" should be "nested grids"; "shows the similar features as revealed with the limited observations" should be something like "shows features similar to those revealed with limited observations"; "favorable" should probably be "favored".*

Thank you. Corrected.

*21. Lines 478-480: These sentences seem largely redundant with the preceding paragraph.*

We have deleted some sentences in the preceding paragraph.

**Reply to Referee # 1**

This manuscript documents the small-scale vortices in the tropical cyclone boundary layer found in a two-way nested WRF-LES set up using the large-scale conditions of a real typhoon. The results are interesting and the presentation is quite clear. I have only a few minor comments about the model setup and interpretation of results.

Minor corrections:

*1. The 100-m vertical resolution is relatively coarse for adequately resolving structures like the high theta_e layers shown in Fig. 7b. It is also coarse compared to the 37m horizontal resolution. A higher vertical resolution is also desirable for capturing the strength and scale of the horizontal roll vortices mentioned by the authors (Fig. 3a). Have the authors done any sensitivity tests to examine the impact of vertical resolution on the structure and distribution of the small-scale vortices focused on in the work?*

We agree with you that the vertical resolution in the innermost domain is relatively coarse compared to the horizontal spacing of 37 m. We did not run experiments to examine the sensitivity to the vertical resolution because of the limit of the computation resource. In fact, we attempted to increase the vertical resolution, but the model cannot run on Tianhe-2 computer. For this reason, we conducted the LES-111 experiment (111.1m horizontal resolution) with 12 vertical levels below 1km. In LES-111 experiment, the vertical resolution and horizontal resolution are comparable in the TC boundary layer. The near-surface linear coherent structures and tornado-scale vortex (TSV) simulated in LES-111 are similar to those in the LES-37 experiment. In the revised manuscript, we have added a brief description about the issue.

*2. I couldn't quite infer the exact connection between the "quasi-linear bands" and what is shown in Fig. 8 (paragraph starting on Line 334). Are the authors implying that the wind speed horizontal variability associated with the quasi-linear features could explain in part the wind speed jump associated with the "tornado-scale vortices"? Please be more explicit.*

Previous studies suggested that the quasi-linear bands are associated with the horizontal rolls in the TC boundary. Our simulation shows that the simulated tornado-scale vortices are closely associated with horizontal rolls inside the RMW. The enhanced vertical motion increases the upward and downward momentum transports (Fig. 7a), amplifying the horizontal gradient of the near-surface wind speed (Fig. 8b). Therefore, the wind speed horizontal variability associated with the quasi-linear features could explain in part the wind speed jump associated with the tornado-scale vortices. The wind speed jump associated with tornado-scale vortices are clear in Fig. 8, but some tornado-scale vortices are not associated with pronounced near-surface wind speed jump. We have made it more explicit in the revised manuscript.

*3. By categorizing the vortices into 3 groups, are the authors suggesting that they are generated/maintained by different physical mechanisms? Could they simply represent different phases in the life cycle of these coherent structures?*

You are right. In this manuscript we did not focus on the mechanisms for tornado-scale vortex generation and maintenance. We think that considerable analysis is needed to understand the mechanisms. While strong vertical and horizontal wind shear inside eyewall may be important for the development of the tornado-scale vortices, we suggest that the three categories of tornado-scale vortices are associated with different hydrostatic stratification.

We used the 3-second model output to examine the evolution of the simulated tornado-scale vortices. It seems that the beginning of most simulated tornado-scale vortices is associated with horizontal rolls. 
[revised manuscript text omitted]

---

## Referee Report (RR1)

**2[nd] Review** of "Tornado-Scale Vortices in the Tropical Cyclone Boundary Layer: Numerical Simulation with the WRF-LES Framework", by Liguang Wu, Qingyuan Liu, and Yubing Li.

**General Comments**:  The authors have satisfactorily addressed most of my comments, and this manuscript has been substantially improved.  I have some remaining minor comments and technical corrections that the authors should address.  I think the most important remaining issue relates to the claimed relationship between the tornado-scale vortices and boundary layer rolls, and I remain unconvinced that the authors have shown such a relationship (and there is some inconsistency in how they discuss this).  I would like the authors to consider revising this aspect of their discussion, but I won't object if they decide not to.  I look forward to seeing the publication of this study.

**Specific Minor Comments**:

1.  P5 l120-123

    It is unclear what the authors mean by referring to track fluctuations here.  I think they are referring to the trochoidal oscillation of the storm center as observed in Hugo by Marks et al. (2008).  However, this feature is distinct from the Eyewall Vorticity Maximum (EVM) that Marks et al. (2008) documented, and Marks et al. (2008) argue that these two phenomena are likely not directly related.  So I don't think there is much evidence to suggest that the track of TCs should be affected by tornado-scale vortices.  The sentence as written is also grammatically incorrect (track fluctuations are implied to be a mechanism for intensification), and it would need to be rewritten.  However, I think it would simplest and most accurate to just remove "and track fluctuations".

2.  P6 l127

    I suggest changing "the numerical simulation" to "a semi-idealized numerical simulation".

3.  P6 l129-134

    The method and motivation of this semi-idealized framework is now much clearer.  I suggest that the authors consider adding a sentence somewhere in this paragraph to emphasize that they are not attempting to simulate a specific real storm.

4.  P7 l147-148

    Are the authors setting the SST to be constant at 29C while maintaining a land surface where it really exists within the domain?  Or are they removing all land from the model domain as well?  This needs to be clarified.

5.  P8 l190

Though there is some evidence of linear structures in Fig. 2b, I'm still not quite convinced that the wind speed field is *dominated* by quasi-linear structures in the eyewall region, and in the zoomed in region in Fig. 3, it doesn't look that dominated by linear structures to me.

The authors remain unclear/contradictory about what they consider to be a roll vortex, and whether they consider the tornado-scale vortices to be manifestations of roll vortices or not. My current belief is that these are distinct phenomena, and while it remains an open question as to whether they are related, I'm unconvinced that the authors have demonstrated any such relationship within this study.

6. P9 l195-196

In my view, I don't think the streaks of high and low wind speeds are as clear as the authors argue that they are. I agree that there are some semi-coherent regions of stronger and weaker winds, but the very strongest winds do *not* seem to me to be oriented in the same way as these more linear features. Since the authors are arguing for the dominance of the quasi-linear features, I suggest that they add a few lines to Fig. 3a to illustrate where they believe that the linear features exist and show their orientation.

7. P9 l201

Please clarify that the statement "the wind speeds generally increase with the increasing window size" is specific to the locations of the vortices. I don't think that it can be true that the perturbation winds increase generally with increasing window size. For example, in a region of the eye with relatively weak absolute wind speed, increasing the window size will include more of the eyewall in the definition of the mean, and so the perturbation wind would decrease there. Whether the perturbation increases or decreases with window size will depend on the variation of the spatial gradient.

8. P10 l217-219

The authors give the magnitude of the updraft in Isabel as 25 m/s, which is the value given in Aberson et al. (2006). However, it is apparent from the figure in Aberson et al. (2006) that this value is somewhat approximate, and not as precise as the 31 m/s given for Felix. The same Isabel sonde is shown in Fig. 4h of Stern and Bryan (2018), and it can be seen that the peak vertical velocity is closer to 22 m/s.

The authors might be interested to note that the strongest dropsonde measured updraft is 27 m/s in Hurricane Patricia (2015) (Rogers et al. 2017).

I suggest giving the approximate altitude instead of the approximate pressure here.

9. P10 l232

   Change "in the 11-hour output" to "at the 11 hourly output times".

10. P12 l274

    Change "east" to "southeast", as a semicircle from the northwest would extend to the southeast.

11. P15 l348-349

    I don't really agree with the characterization of the updraft-downdraft couplet as a horizontal roll vortex. Although this feature possesses horizontal vorticity, the term "roll vortex" is generally used to distinguish quasi-linear features, and the tornado-scale vortices shown in this study do not have such a structure.

12. P16 l364-36

    I'm not sure that the phrase "due to" is correct here. Studies have associated quasi-linear bands with horizontal rolls, and these rolls are characterized by a transverse circulation with alternating upward and downward motion. But the rolls are not "due to" (i.e., caused by) the momentum transports. I think this might simply be imprecise language. For clarity, rewrite as "Some previous studies….in the TC boundary layer, with alternating upward and downward momentum transport on either side of the rolls".

13. P16 l367-368

    It is unclear to me how Fig. 8b demonstrates a relationship of the tornado-scale vortices with boundary layer rolls, as the authors claim here. That the winds increase sharply across the vortices is consistent with this being a vortex, but don't demonstrate anything about whether these features are related to quasi-linear boundary layer roles.

    Change "cross section of winds" to "radial profile of winds", as this is not a two-dimensional cross section.

14. P17 l382

    The authors refer to vertical motion of >12 m/s extending to 2 km height. This doesn't seem correct to me; it looks like perhaps the 8 m/s contour extends this high.

    Also, the authors still need to clarify here whether the values of vertical velocity (here and elsewhere) refer to the total or a perturbation value. It appears that this must be a perturbation

vertical velocity, because otherwise these magnitudes wouldn't satisfy the 20 m/s threshold for vortex identification that the authors are using. The use of the perturbation vertical velocity may be unclear to the reader, and so the authors need to be explicit.

15. P19 l428-429

This sentence is repetitive with the content of the previous paragraph, and it also just gives the chosen definition, not an independent result, so I would suggest removing it.

16. Fig. 1

Clarify both here and in the text that *both* of these measures of the wind speed are instantaneous. As written, it is implied that there is a contrast between the "instantaneous" winds and the "azimuthal-mean" winds, but the azimuthal-mean winds are also (as far as I can tell) instantaneous. The distinction between them is that the red curve shows the local (point-value) maximum, and the blue line shows the maximum of the azimuthally averaged wind.

17. Fig. 2

The box shown here is actually 80x80 km, not 40x40 km as stated. The units should be "km", not "km$^2$". The font size on l654 appears to be different from the rest of the caption. Remove "(27h)" from l656.

**Technical Corrections**:

1. P2 l50-51, and elsewhere

   "rolling vortices" should be "roll vortices".

2. P6 l136

   I think it would clearer if "centered at 30.0N, 132.5E" were placed within parentheses.

3. P6 l145, and elsewhere

   Where domain sizes are given such as here, the proper units should be "km", not "km$^2$", because this is expressing a length scale, not an area. So this should be written as "90x90 km", and similarly throughout the manuscript.

4. P7 l164, and elsewhere

I suggest changing "figure not shown" to just "not shown", as that is the more conventional usage (it's also fine to keep this as is, if the authors prefer).

5. P8 l176, and elsewhere

This is actually a 10-hour period, *not* an 11-hour period, as it is the length of time between t=26 h and t=36 h.  Though there are 11 output times, the period is 10 hours.  So "11-hour period" should be changed to "10-hour period" throughout the manuscript.

6. P8 l187

Change "landfall on Florida" to "landfall in Florida", as this is the conventional expression.

7. P9 l193

"feature" should be "features".

8. P9 l193

For consistency with convention, I think "7x10 km" should be written as "10x7 km", to give the zonal (x) dimension first.

9. P9 l199

Change "compare" to "compared".

10. P9 l203-204

For clarity, rewrite this sentence to say "The simulated small-scale circulations are similar to those found from instead calculating the perturbations by subtracting the symmetric and wavenumber 1-3 components with respect to the TC center (not shown)."

11. P9 l209-210

Change "Compared to Figure 3a" to "Comparing Figs. 3a and 3b, it can be seen that".

12. P9 l214

"boundary" should be "boundary layer".

13. P10 l216

Change "the small scale vortex" to "a small scale vortex".

14. P10 l222

"treat" should be "treated".  Insert "a" prior to "tornado-scale".

15. P11 l258

Is this really the "30th hour"?  Or is it starting from t=30 h?  These are different, as the 30th hour would be t=29-30 h.

16. P12 l264

Change "1-hour" to "hourly".  Remove the word "Besides", and start the sentence with "The durations of…"

17. P12 l278

Suggest changing "the two" to "these two".

18. P14 l327-328

Change "its vertical" to "their vertical" (in two places).  "extension" should be "extent".

19. P17 l383

"the near surface" should be "near the surface".

20. P18 l399-400

Change "seems to be" to "is".
There is a missing space after the period after "1km."

21. P18 l400

Remove "the" after "conducted".

22. P18 l401

Insert "(not shown)" after "an experiment".  Change "In the experiment" to "In this experiment".

23. P18 l402

   Insert "the" before "TC". Combine the two sentences on lines 402-403, such as "In this experiment, the vertical and horizontal resolutions are comparable in the boundary layer, and the tornado-scale vortices can still be found.

24. P18 l404

   Change "that the" to "if these".

25. P18 l405

   Insert "and" prior to "since".

26. P18 l413

   I think conventionally, this simulation would be considered to have "seven nested grids", not six, as the parent domain is itself generally included in the total.

27. P19 l420

   Change "11-hour" to "11 hours of".

28. P19 l430

   "Nearly in all" should be "In nearly all".

29. Fig. 6

   Change "cycle" to "oval".

---

## Author Response (AR2)

**General Comments:**

The authors have satisfactorily addressed most of my comments, and this manuscript has been substantially improved. I have some remaining minor comments and technical corrections that the authors should address. I think the most important remaining issue relates to the claimed relationship between the tornado-scale vortices and boundary layer rolls, and I remain unconvinced that the authors have shown such a relationship (and there is some inconsistency in how they discuss this). I would like the authors to consider revising this aspect of their discussion, but I won't object if they decide not to. I look forward to seeing the publication of this study.

Specific Minor Comments:

1. *P5 |120-123*

   *It is unclear what the authors mean by referring to track fluctuations here. I think they are referring to the trochoidal oscillation of the storm center as observed in Hugo by Marks et al. (2008). However, this feature is distinct from the Eyewall Vorticity Maximum (EVM) that Marks et al. (2008) documented, and Marks et al. (2008) argue that these two phenomena are likely not directly related. So I don't think there is much evidence to suggest that the track of TCs should be affected by tornado-scale vortices. The sentence as written is also grammatically incorrect (track fluctuations are implied to be a mechanism for intensification), and it would need to be rewritten. However, I think it would simplest and most accurate to just remove "and track fluctuations".*

   Removed.

2. *P6 |127*

   *I suggest changing "the numerical simulation" to "a semi-idealized numerical simulation".*

   Changed.

3. *P6 |19-134*

*The method and motivation of this semi-idealized framework is now much clearer. I suggest that the authors consider adding a sentence somewhere in this paragraph to emphasize that they are not attempting to simulate a specific real storm.*

We have made it clear in the revised manuscript..

4. *P7/147-148*

   *Are the authors setting the SST to be constant at 29C while maintaining a land surface where it really exists within the domain? Or are they removing all land from the model domain as well? This needs to be clarified. Are the authors setting the SST to be constant at 29C while maintaining a land surface where it really exists within the domain? Or are they removing all land from the model domain as well? This needs to be clarified.*

   In this study, all of the land was removed and the SST was set to be a constant of 29°C.

5. *P8/190*

   *Though there is some evidence of linear structures in Fig. 2b, I'm still not quite convinced that the wind speed field is dominated by quasi-linear structures in the eyewall region, and in the zoomed in region in Fig. 3, it doesn't look that dominated by linear structures to me.*

   *The authors remain unclear/contradictory about what they consider to be a roll vortex, and whether they consider the tornado-scale vortices to be manifestations of roll vortices or not. My current belief is that these are distinct phenomena, and while it remains an open question as to whether they are related, I'm unconvinced that the authors have demonstrated any such relationship within this study.*

   We agree with you that the tornado-scale vortices are distinct phenomena from the roll vortex. Since the near surface quasi-linear structures are coupled with the tornado-scale vortex from Fig. 6, we think that the formation of the tornadoscale vortex may be related to the roll vortex. The manuscript has been revised to make it clearer.

6. *P9/195-196*

*In my view, I don't think the streaks of high and low wind speeds are as clear as the authors argue that they are. I agree that there are some semi-coherent regions of stronger and weaker winds, but the very strongest winds do not seem to me to be oriented in the same way as these more linear features. Since the authors are arguing for the dominance of the quasi-linear features, I suggest that they add a few lines to Fig. 3a to illustrate where they believe that the linear features exist and show their orientation.*

You are right. The strongest winds associated with the tornado-scale vortex do not to be oriented in the same way as the quasi-linear structure.

7. *P9/201*

*Please clarify that the statement "the wind speeds generally increase with the increasing window size" is specific to the locations of the vortices. I don't think that it can be true that the perturbation winds increase generally with increasing window size. For example, in a region of the eye with relatively weak absolute wind speed, increasing the window size will include more of the eyewall in the definition of the mean, and so the perturbation wind would decrease there. Whether the perturbation increases or decreases with window size will depend on the variation of the spatial gradient.*

The statement means that the *maximum* perturbation wind speed generally increases with the increase of the window size of the running smooth. We find that the increase is slow when the window size is larger than 6 km. Therefore, we use the window size of 8 km. We have revised the statement.

8. *P10/217-219*

*The authors give the magnitude of the updraft in Isabel as 25 m/s, which is the value given in Aberson et al. (2006). However, it is apparent from the figure in Aberson et al. (2006) that this value is somewhat approximate, and not as precise as the 31 m/s given for Felix. The same Isabel sonde is shown in Fig. 4h of Stern and Bryan (2018), and it can be seen that the peak vertical velocity is closer to 22 m/s.*

*The authors might be interested to note that the strongest dropsonde measured updraft is 27 m/s in Hurricane Patricia (2015) (Rogers et al. 2017).*

*I suggest giving the approximate altitude instead of the approximate pressure here.*

Thank you for your suggestions.

We have checked Fig. 4h of Stern and Bryan (2018). The updraft of 22 m s$^{-1}$ in Isabel was detected by a GPS dropwindsonde just at about 1300 m. We have revised the description in our manuscript.

9. *P10/232*

*Change "in the 11-hour output" to "at the 11 hourly output times".*

Changed.

10. *P12/274*

*Change "east" to "southeast", as a semicircle from the northwest would extend to the southeast.*

Changed.

11. *P15/348-349*

*I don't really agree with the characterization of the updraft-downdraft couplet as a horizontal roll vortex. Although this feature possesses horizontal vorticity, the term "roll vortex" is generally used to distinguish quasi-linear features, and the tornado-scale vortices shown in this study do not have such a structure.*

Thanks for your suggestion. The "horizontal rolling vortex" has replaced by "vertical circulation".

12. *P16/364-365*

*I'm not sure that the phrase "due to" is correct here. Studies have associated quasi-linear bands with horizontal rolls, and these rolls are characterized by a transverse circulation with alternating upward and downward motion. But the rolls are not "due to" (i.e., caused by) the momentum transports. I think this might simply be imprecise language. For clarity, rewrite as "Some previous studies....in the TC boundary layer, with alternating upward and downward momentum transport on either side of the rolls".*

Thank you. Revised.

13. *P16/367-368*

*It is unclear to me how Fig. 8b demonstrates a relationship of the tornado-scale vortices with boundary layer rolls, as the authors claim here. That the winds increase sharply across the vortices is consistent with this being a vortex, but don't demonstrate anything about whether these features are related to quasi-linear boundary layer rolls.*

Please see our No. 6 response.

*Change "cross section of winds" to "radial profile of winds", as this is not a two-dimensional cross section.*

Changed.

14. *P17/382*

   *The authors refer to vertical motion of >12 m/s extending to 2 km height. This doesn't seem correct to me; it looks like perhaps the 8 m/s contour extends this high.*

   Revised.

   *Also, the authors still need to clarify here whether the values of vertical velocity (here and elsewhere) refer to the total or a perturbation value. It appears that this must be a perturbation vertical velocity, because otherwise these magnitudes wouldn't satisfy the 20 m/s threshold for vortex identification that the authors are using. The use of the perturbation vertical velocity may be unclear to the reader, and so the authors need to be explicit.*

   The values of vertical velocity refer to perturbation values.

   Revised.

15. *P19/428-429*

   *This sentence is repetitive with the content of the previous paragraph, and it also just gives the chosen definition, not an independent result, so I would suggest removing it.*

   Removed.

16. *Fig. 1*

   *Clarify both here and in the text that both of these measures of the wind speed are instantaneous. As written, it is implied that there is a contrast between the "instantaneous" winds and the "azimuthal-mean" winds, but the azimuthal-mean winds are also (as far as I can tell) instantaneous. The distinction between them is that the red curve shows the local (pointvalue) maximum, and the blue line shows the maximum of the azimuthally averaged wind.*

   Added.

17. *Fig. 2*

*The box shown here is actually 80x80 km, not 40x40 km as stated.    The units should be "km", not "km2".    The font size on l654 appears to be different from the rest of the caption. Remove "(27h)" from l656.*

Thank you. Revised.

**Technical Corrections:**

1.  P2 |50-51, and elsewhere

    *"rolling vortices" should be "roll vortices".*

    Revised.

2.  P6 |136

    *I think it would clearer if "centered at 30.0N, 132.5E" were placed within parentheses.*

    Changed.

3.  P6 |145, and elsewhere

    *Where domain sizes are given such as here, the proper units should be "km", not "km2", because this is expressing a length scale, not an area.    So this should be written as "90x90 km", and similarly throughout the manuscript.*

    Revised.

4.  P7 |164, and elsewhere

    *I suggest changing "figure not shown" to just "not shown", as that is the more conventional usage (it's also fine to keep this as is, if the authors prefer).*

    Revised.

5.  P8 |176, and elsewhere

    *This is actually a 10-hour period, not an 11-hour period, as it is the length of time between t=26 h and t=36 h.    Though there are 11 output times, the period*

*is 10 hours. So "11-hour period" should be changed to "10-hour period" throughout the manuscript.*

Changed.

6.  P8 |187

    *Change "landfall on Florida" to "landfall in Florida", as this is the conventional expression.*

    Changed.

7.  P9 |193

    *"feature" should be "features".*

    Revised.

8.  P9 |193

    *For consistency with convention, I think "7x10 km" should be written as "10x7 km", to give the zonal (x) dimension first.*

    Revised.

9.  P9 l199

    *Change "compare" to "compared".*

    Changed.

10. P9 |203-204

    *For clarity, rewrite this sentence to say "The simulated small-scale circulations are similar to those found from instead calculating the perturbations by subtracting the symmetric and wavenumber 1-3 components with respect to the TC center (not shown)."*

    Thank you. The sentence is rewritten.

11. P9 |209-210

    *Change "Compared to Figure 3a" to "Comparing Figs. 3a and 3b, it can be seen that".*

    Changed.

12. P9 |214

    *"boundary" should be "boundary layer".*

Changed.

13. P10 |216

*Change "the small scale vortex" to "a small scale vortex".*

Changed.

14. P10 |222

*"treat" should be "treated".   Insert "a" prior to "tornado-scale".*

Changed.

15. P11 |258

*Is this really the "30th hour"?   Or is it starting from t=30 h?   These are different, as the 30th hour would be t=29-30 h.*

"30th hour" is replaced by "31th hour".

16. P12 |264

*Change "1-hour" to "hourly".   Remove the word "Besides", and start the sentence with "The durations of…"*

Revised.

17. P12 |278

*Suggest changing "the two" to "these two".*

Changed.

18. P14 |327-328

*Change "its vertical" to "their vertical" (in two places).   "extension" should be "extent".*

Changed.

19. P17 |383

*"the near surface" should be "near the surface".*

Replaced.

20. P18 |399-400

*Change "seems to be" to "is".*

*There is a missing space after the period after "1km."*

Revised.

21. P18 |400

*Remove "the" after "conducted".*

Removed.

22. P18 |401

*Insert "(not shown)" after "an experiment".   Change "In the experiment" to "In this experiment".*

Inserted and changed.

23. P18 |402

*Insert "the" before "TC". Combine the two sentences on lines 402-403, such as "In this experiment, the vertical and horizontal resolutions are comparable in the boundary layer, and the tornado-scale vortices can still be found.*

Revised.

24. P18 |404

*Change "that the" to "if these".*

Changed.

25. P18 |405

*Insert "and" prior to "since".*

Inserted.

26. P18 |413

*I think conventionally, this simulation would be considered to have "seven nested grids", not six, as the parent domain is itself generally included in the total.*

We think the parent domain is not included in the nested grids.

27. P19 |420

*Change "11-hour" to "11 hours of".*

Changed.

28. P19 |430

   *"Nearly in all" should be "In nearly all".*

   Changed.

29. Fig. 6

   *Change "cycle" to "oval".*

   Changed.